# Dynamic Model Predictive Shielding for Provably Safe Reinforcement Learning

**Arko Banerjee**
The University of Texas at Austin
arko.banerjee@utexas.edu

**Kia Rahmani**
The University of Texas at Austin
kia@durable.ai

**Joydeep Biswas**
The University of Texas at Austin
joydeepb@utexas.edu

**Isil Dillig**
The University of Texas at Austin
isil@utexas.edu

## Abstract

Among approaches for provably safe reinforcement learning, Model Predictive Shielding (MPS) has proven effective at complex tasks in continuous, high-dimensional state spaces, by leveraging a *backup policy* to ensure safety when the learned policy attempts to take unsafe actions. However, while MPS can ensure safety both during and after training, it often hinders task progress due to the conservative and task-oblivious nature of backup policies. This paper introduces Dynamic Model Predictive Shielding (DMPS), which optimizes reinforcement learning objectives while maintaining provable safety. DMPS employs a local planner to dynamically select safe recovery actions that maximize both short-term progress as well as long-term rewards. Crucially, the planner and the neural policy play a synergistic role in DMPS. When planning recovery actions for ensuring safety, the planner utilizes the neural policy to estimate long-term rewards, allowing it to *observe* beyond its short-term planning horizon. Conversely, the neural policy under training learns from the recovery plans proposed by the planner, converging to policies that are both *high-performing* and *safe* in practice. This approach guarantees safety during and after training, with bounded recovery regret that decreases exponentially with planning horizon depth. Experimental results demonstrate that DMPS converges to policies that rarely require shield interventions after training and achieve higher rewards compared to several state-of-the-art baselines.

## 1   Introduction

Safe Reinforcement Learning (SRL) [1, 2] aims to learn policies that adhere to important safety requirements and is essential for applying reinforcement learning to safety-critical applications, such as autonomous driving. Some SRL approaches give *statistical* guarantees and reduce safety violations by directly incorporating safety constraints into the learning objective [3, 4, 5, 6]. In contrast, *Provably Safe* RL (PSRL) methods aim to learn policies that *never* violate safety and are essential in high-stakes domains where even a single safety violation can be catastrophic [7].

A common approach to PSRL is *shielding* [8], where actions proposed by the policy are monitored for potential safety violations. If necessary, these actions are overridden by a safe action that is guaranteed to retain the system's safety. Model Predictive Shielding (MPS) is an emerging PSRL method that has proven effective in high-dimensional, continuous state spaces with complex dynamics, surpassing previous shielding approaches in applicability and performance [9, 10]. At a high level, MPS methods leverage the concept of *recoverable states* that lead to a safe equilibrium in $N$ time-steps by following

38th Conference on Neural Information Processing Systems (NeurIPS 2024).

a so-called *backup policy*. MPS approaches dynamically check (via simulation) if a proposed action results in a recoverable state and follow the backup policy if it does not.

However, an important issue with existing MPS approaches is that the backup policies are not necessarily aligned with the primary task's objectives and often propose actions that, while safe, impede progress towards task completion—even when alternative safe actions are available that do not obstruct progress in the task. Intuitively, this occurs because the backup policy is designed with the sole objective of driving the agent to the closest equilibrium point rather than making task-specific progress. For instance, in an autonomous navigation task, if the trained policy suggests an action that could result in a collision, MPS methods revert to a backup policy that simply instructs halting, rather than trying to find a more optimal recovery approach, such as finding a route that steers around the obstacle. As a result, the recovery phase of MPS frequently inhibits the learning process and incurs high *recovery regret*, meaning that there is a large discrepancy between the return from executed recovery actions and the maximum possible returns from the same states.

Motivated by this problem, we propose a novel PSRL approach called *Dynamic Model Predictive Shielding* (DMPS), which aims to optimize RL objectives while ensuring provable safety under complex dynamics. The key idea behind DMPS is to employ a local planner [11, 12] to dynamically identify a safe recovery action that optimizes finite-horizon progress towards the goal. Although the computational overhead of planning grows exponentially with the depth of the horizon, in practice, a reasonably small horizon is sufficient for allowing the agent to recover from unsafe regions.

In DMPS, the planner and the neural policy under training play a synergistic role. First, when using the planner to recover from potentially unsafe regions, our optimization objective not only uses the finite-horizon reward but also uses the Q-function learned by the neural policy. The integration of the Q-function into the optimization objective allows the planner to take into account *long-term reward* beyond the short planning horizon needed for recovering safety. As a result, the planner benefits from the neural policy in finding recovery actions that optimize for long-term reward. Conversely, the integration of the planner into the training loop allows the neural policy to learn from actions suggested by the planner: Because the planner dynamically figures out how to avoid unsafe regions while making task-specific progress, the policy under training also learns how to avoid unsafe regions in a *smart* way, rather than taking overly conservative actions. From a theoretical perspective, DMPS can guarantee provable safety both during and after training. We also provide a theoretical guarantee that the regret from recovery trajectories identified by DMPS is bounded and that it decays at an exponential rate with respect to the depth of the planning horizon. These properties allow DMPS to learn policies that are both high-performing and safe.

We have implemented the DMPS algorithm in an open-source library and evaluated it on a suite of 13 representative benchmarks. We experimentally compare our approach against Constrained Policy Optimization (CPO) [3], PPO-Lagrangian (PPO-Lag) [13, 14], Twin Delayed Deep Deterministic Policy Gradient (TD3) [15] and state-of-the-art PSRL approaches, MPS [9] and REVEL [16]. Our results indicate that policies learned by DMPS outperform all baselines in terms of total episodic return and achieve safety with minimal shield invocations. Specifically, DMPS invokes the shield 76% less frequently compared to the next best baseline, MPS, and achieves 29% higher returns after convergence.

To summarize, the contributions of this paper are as follows: First, we introduce the DMPS algorithm, a novel integration of RL and planning, that aims to address the limitations of model predictive shielding. Second, we provide a theoretical analysis of our algorithm and prove that recovery regret decreases exponentially with planning horizon depth. Lastly, we present a detailed empirical evaluation of our approach on a suite of PSRL benchmarks, demonstrating superior performance compared to several state-of-the-art baselines.

## 2 Related Work

***PSRL.*** There is a growing body of work addressing safety issues in RL [1, 2, 17, 18, 19, 20, 21]. Our approach falls into the category of provably safe RL (PSRL) techniques [7, 18], treating safety as a hard constraint that must never be violated. This is in contrast to *statistically safe* RL techniques, which provide only statistical bounds on the system's safety by constraining the training objectives [3, 22, 23, 24, 25]. These soft guarantees, however, are insufficient for domains like autonomous driving, where each failure can be catastrophic. Existing PSRL methods can be categorized based on whether

they guarantee safety *throughout the training phase* [26, 27, 16, 28] or only *post-training* upon deployment [29, 30, 31, 32, 33]. Our approach falls into the former category and employs a shielding mechanism that ensures safety both during training and deployment.

***Safety Shielding.*** Many PSRL works use *safety shields* [8, 34, 35, 36, 37, 16, 38, 39]. These methods typically synthesize a domain-specific policy ahead of time and use it to detect and substitute unsafe actions at runtime. However, traditional shielding methods tend to be computationally intractable, leading to limited usability. For instance, [8] presents one of the early works in shielding, where safety constraints are specified in Linear Temporal Logic (LTL), and a verified reactive controller is synthesized to act as the safety shield at runtime. However, this approach is restricted to discrete state and action spaces, due to the complexity of shield synthesis. Another example is [39], which enhances PSRL performance by substituting shield-triggering actions with verified actions and reducing the frequency of shield invocations. In Model Predictive Shielding (MPS) [9], a backup policy dynamically assesses the states from which safety can be reinstated and, if necessary, proposes substitute actions. Unlike pre-computed shields, MPS can handle high-dimensional, non-linear systems with continuous states and actions, without incurring an exponential runtime overhead [40]. MPS has also been successfully applied to stochastic dynamics systems [41] and multi-agent environments [10]. However, a significant limitation of current MPS methods is the separation of safety considerations from the RL objectives, which hinders learning when employing the recovery policy.

***RL and Planning.*** Planning has traditionally been considered a viable complement to reinforcement learning, combining real-time operations in both the actual world and the learned environment model [42, 43, 44, 45, 46]. With recent developments in deep model-based RL [44, 47, 48, 49, 50] and the success of planning algorithms in discrete [51, 52] and continuous spaces [53, 54, 55, 56, 57], the prospect of combining these methods holds great promise for solving challenging real-world problems. Integrating planning within RL has also been applied to safety measures [58, 59, 21, 60, 61, 62, 63, 64]. For instance, Safe Model-Based Policy Optimization [21] minimizes safety violations by detecting non-recoverable states through forward planning using an accurate dynamics model of the system. However, it only employs planning to identify unsafe states, not to find optimal recovery paths from such situations. To the best of our knowledge, `DMPS` is the first method that leverages dynamic planning for optimal recovery, within the framework of model predictive shielding.

***Classical Control.*** There is a long line of classical control approaches to the problem of safe navigation [65, 66, 67, 68, 69, 70]. One common approach, for instance, is the use of *control barrier functions* (CBFs) [71, 72]. CBFs have been used across many domains in robotics to achieve safety with high confidence [73, 74, 75, 76]. While useful, these approaches tend to make assumptions about the environment (e.g. differentiability, closed-form access to dynamics) that cannot easily be reconciled with the highly general RL framing of this work.

## 3   Preliminaries

***MDP.*** We formulate our problem using a standard Markov Decision Process (MDP) $\mathcal{M} = \langle \mathcal{S}, \mathcal{S}_U, \mathcal{S}_0, \mathcal{A}, \mathcal{T}, \mathcal{R}, \gamma \rangle$, where $\mathcal{S} \subseteq \mathbb{R}^n$ is a set of states, $\mathcal{S}_U \subset \mathcal{S}$ is a set of unsafe states, $\mathcal{S}_0 \subset \mathcal{S}$ are the initial states, $\mathcal{A} \subseteq \mathbb{R}^m$ is a continuous action space, $\mathcal{T} : \mathcal{S} \times \mathcal{A} \to \mathcal{S}$ is a deterministic transition function, $\mathcal{R} : \mathcal{S} \times \mathcal{A} \to \mathbb{R}$ is a deterministic reward function, and $\gamma$ is the discount factor. We define $\Pi$ to be the set of all agent policies, where each policy $\pi \in \Pi$ is a function from environment states to actions, *i.e.,* $\pi : \mathcal{S} \to \mathcal{A}$. For any set $S \subseteq \mathcal{S}$, the set of *reachable states* from $S$ in $i$ steps under policy $\pi$, is denoted by $\texttt{reach}_i(\pi, S)$, and is defined recursively as follows: $\texttt{reach}_1(\pi, S) \doteq \{\mathcal{T}(s, \pi(s)) \mid s \in S\}$ and $\texttt{reach}_{i+1}(\pi, S) \doteq \texttt{reach}_1(\pi, \texttt{reach}_i(\pi, S))$. The set of *all* reachable states under a policy $\pi$ is $\texttt{reach}(\pi) \doteq \bigcup_{1 \le i} \texttt{reach}_i(\pi, \mathcal{S}_0)$.

***PSRL.*** The standard objective in RL is to find a policy $\pi$ that maximizes a performance measure, $J(\pi)$, typically defined as the expected infinite-horizon discounted total return. In provably safe reinforcement learning (PSRL), the aim is to identify a *safe* policy that maximizes the above measure. The set of safe policies is denoted by $\Pi_{safe} \subseteq \Pi$ and consists of policies that never reach an unsafe state, *i.e.,* $\pi \in \Pi_{safe} \Leftrightarrow \texttt{reach}(\pi) \cap \mathcal{S}_U = \emptyset$. Therefore, the objective of PSRL is to find a policy $\pi_{safe}^* \doteq \arg\max_{\pi \in \Pi_{safe}} J(\pi)$.

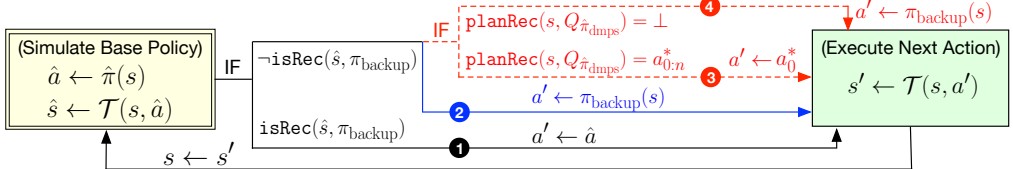

Figure 1: Overview of an execution cycle in MPS (❶, ❷) and DMPS (❶, ❷, ❸, ❹).

## 4 Model Predictive Shielding

Under the Model Predictive Shielding (MPS) framework [9], a safe policy, $\pi_{\mathrm{mps}}$, is constructed by integrating two sub-policies: a *learned policy*, $\hat{\pi}_{\mathrm{mps}}$, and a *backup policy*, $\pi_{\mathrm{backup}}$. Depending on the current state of the system, control of the agent's actions is delegated to one of these two policies. The learned policy is typically implemented as a neural network that is trained using standard deep RL techniques to optimize $J(\cdot)$. However, this policy may violate safety during training or deployment, *i.e.,* $\hat{\pi}_{\mathrm{mps}} \notin \Pi_{\mathrm{safe}}$. On the other hand, the backup policy, $\pi_{\mathrm{backup}}$, is specifically designed for safety and is invoked to substitute potentially unsafe actions proposed by the learned policy.

Due to domain-specific constraints on system transitions, the backup policy is effective only from a certain subset of states, called *recoverable* states. For instance, given the deceleration limits of an agent, a backup policy that instructs the agent to halt can only avoid collisions from states where there is sufficient distance between the agent and the obstacle. At a high level, the recoverability of a given state $s$ in MPS is determined by a function $\mathtt{isRec} : \mathcal{S} \times \Pi \rightarrow \mathbb{B}$. This function performs an $N$-step forward simulation of $\pi_{\mathrm{backup}}$ from $s$ and checks whether a safety equilibrium can be established.

Figure 1 gives an overview of the control delegation mechanism in $\pi_{\mathrm{mps}}$.[1] Given state $s \in \mathcal{S}$, the agent first forecasts the next state $\hat{s}$ that would be reached by following the learned policy (double-bordered, yellow box). If $\mathtt{isRec}(\hat{s}, \pi_{\mathrm{backup}})$ is true, then $\pi_{\mathrm{mps}}$ simply returns the same action as $\hat{\pi}_{\mathrm{mps}}$, as marked by ❶. Otherwise, if $\mathtt{isRec}(\hat{s}, \pi_{\mathrm{backup}})$ is false, $\pi_{\mathrm{mps}}$ delegates control to the backup policy $\pi_{\mathrm{backup}}$, as indicated by ❷. The selected action, $a'$, is then executed in the environment (single-bordered, green box), resulting in a new state $s'$. From this state, $\pi_{\mathrm{mps}}$ is executed again, and the process repeats.

The safety of $\pi_{\mathrm{mps}}$ relies on the fact that all recoverable states are safe and that the backup policy is *closed* under the set of recoverable states. Thus, we can inductively show that the agent remains in safe and recoverable states; thus $\pi_{\mathrm{mps}} \in \Pi_{safe}$.

### *Example.*

Consider an agent on a 2D plane aiming to reach a goal while avoiding static obstacles. Figure 2 (a) presents an unsafe trajectory proposed by the learned policy. Figure 2 (b) presents the trajectory under MPS. As discussed above, $\pi_{\mathrm{mps}}$ delegates control to the backup policy in such potentially unsafe situations, which corresponds here to applying maximum deceleration away from the obstacle. This causes the agent to come to a halt, which is suboptimal. Figure 2 (c) depicts a DMPS trajectory. Figure 2 (d) depicts DMPS planning in an environment containing a static obstacle and a low-reward puddle region. The latter two are explained in section 5.

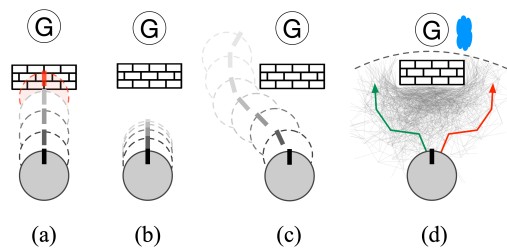

Figure 2: (a) Unsafe trajectory leading to a collision. (b) Safe but sub-optimal trajectory. (c) Optimal and safe trajectory. (d) An instance of the planning phase.

### 4.1 Recovery Regret

While MPS guarantees worst-case safety, shield interventions can hinder the training of $\hat{\pi}_{\mathrm{mps}}$ and compromise overall efficacy. To formalize this limitation, we introduce the concept of *Recovery*

---

[1]Arrows ❸ and ❹ are used for planner-guided recovery, and are explained more in section 5.

*Regret*, which measures the expected performance gap between $\pi_\text{backup}$ and the optimal policy. To this end, we first introduce a helper function $\mathbb{I}_\text{backup} : \mathcal{S} \to \{0, 1\}$, which, given a state $s$, indicates whether control in $s$ is delegated to the backup policy, *i.e.,* $\mathbb{I}_\text{backup}(s) = 1$ if $\neg\texttt{isRec}(\mathcal{T}(s, \hat{\pi}_\text{mps}(s), \pi_\text{backup}))$ and $\mathbb{I}_\text{backup}(s) = 0$ otherwise. In any state $s$ where control is delegated to the backup policy, the *optimal value* $V^*(s)$ represents the maximum expected reward that can be achieved from $s$, and the *optimal action-value* $Q^*(s, \pi_\text{backup}(s))$ represents the value of executing $\pi_\text{backup}$ in $s$ and thereafter following an optimal policy. Thus, we can define Recovery Regret ($RR$) as the expected discrepancy between these two values across states where $\pi_\text{backup}$ is invoked, *i.e.,*

$$RR(\pi_\text{backup}, \pi_\text{mps}) \doteq \mathbb{E}_{s \sim \rho^{\pi_\text{mps}}} \left[ \mathbb{I}_\text{backup}(s) \cdot \left[ V^*(s) - Q^*(s, \pi_\text{backup}(s)) \right] \right]$$

In the above formula, $\rho^{\pi_\text{mps}}$ is the *discounted state visitation distribution* associated with $\pi_\text{mps}$, *i.e.,* $\rho^{\pi_\text{mps}}(s) \doteq (1 - \gamma) \sum_{t=0}^{\infty} \gamma^t Pr(s_t = s \mid s_0 \in \mathcal{S}_0, \; \forall_{i \geq 0}. \; s_{i+1} = \mathcal{T}(s_i, \pi_\text{mps}(s_i)))$. This term assists in quantifying how frequently each state is visited when measuring the regret of a backup policy.

In existing MPS approaches, the misalignment between backup policies and the optimal policy can result in substantial recovery regrets. For example, Figure 2 (c) illustrates the optimal sequence of actions in the previously discussed scenario, where the agent avoids a collision by maneuvering around the obstacle. However, in MPS, the training process lacks mechanisms to teach such optimal behavior to $\hat{\pi}_\text{mps}$ and only teaches overly cautious and poorly rewarded actions depicted in Figure 2 (b).

## 5 Dynamic Model Predictive Shielding

In this section, we introduce the Dynamic Model Predictive Shielding (DMPS) framework that builds on top of MPS but aims to address its shortcomings. Similar to the control delegation mechanism in MPS, the policy $\pi_\text{dmps}$ initially attempts to select its next action using a learned policy $\hat{\pi}_\text{dmps}$. If the action proposed by $\hat{\pi}_\text{dmps}$ leads to a state that is not recoverable using $\pi_\text{backup}$, an alternative safe action is executed instead, as in standard MPS. However, rather than using the task-oblivious policy $\pi_\text{backup}$, the backup action is selected using a *dynamic backup policy* $\pi_\text{backup}^*$. More formally,

$$\pi_\text{dmps}(s) = \begin{cases} \hat{\pi}_\text{dmps}(s) & \text{if } \texttt{isRec}(\mathcal{T}(s, \hat{\pi}_\text{dmps}(s)), \pi_\text{backup}) \\ \pi_\text{backup}^*(s) & \text{otherwise} \end{cases} \tag{1}$$

The core innovation behind DMPS is the dynamic backup policy $\pi_\text{backup}^*$, which is realized using a *local planner* [45, 77], denoted as a function $\texttt{planRec}()$. At a high level, $\texttt{planRec}$ performs a finite-horizon lookahead search over a predetermined number of steps ($n \in \mathbb{N}$) and identifies a sequence of backup actions optimizing the expected returns during recovery. Specifically, the function $\texttt{planRec}$ takes an initial state $s_0$ and a state-action value function $Q : \mathcal{S} \times \mathcal{A} \to \mathbb{R}$, and returns a sequence of task-optimal actions $a_{0:n}^* \in \mathcal{A}^{n+1}$ defined as follows:

$$\texttt{planRec}(s_0, Q) \doteq \underset{a_{0:n} \in \mathcal{A}^{n+1}}{\arg\max} \left[ \left( \sum_{i=0}^{n-1} \gamma^i \cdot \mathcal{R}(s_i, a_i) \right) + \gamma^n \cdot Q(s_n, a_n) \right],$$
$$\text{such that, } \forall_{0 \leq i < n}[s_{i+1} = \mathcal{T}(s_i, a_i)] \text{ and } \forall_{0 \leq i \leq n} \texttt{isRec}(s_i, \pi_\text{backup}). \tag{2}$$

Crucially, the recovery plan is required to only lead to *recoverable* states within the finite planning horizon (*i.e.,* $\texttt{isRec}(s_i, \pi_\text{backup})$). Note that the backup policy $\pi_\text{backup}$ is used by the planner in deciding the recoverability of a state. Beyond satisfying the hard safety constraint, the planner is also required to optimize the objective function shown in Equation 2. Importantly, this objective accounts for both the immediate rewards within the planning horizon, $\mathcal{R}(s_i, a_i)$, *as well as* the estimated long-term value from the terminal state $s_n$ beyond the planning horizon, as defined by $Q$. As a result, the planner benefits from the long-term reward estimates learned by the neural policy $\hat{\pi}_\text{dmps}$.

Figure 2 (d) illustrates an agent planning a recovery path around an obstacle. The actions considered by $\texttt{planRec}$ are represented by gray edges. Two optimal paths on this tree, depicted in green and red, yield similar rewards due to the symmetric nature of the reward function relative to the goal position. The planner selects the green path on the left as its final choice due to a water puddle on the right side of the goal, which, if traversed, would result in lower returns. Since the puddle lies outside of the planning horizon, the decision to opt for the green path is informed by access to the

---

**Algorithm 1**: Reinforcement Learning with Dynamic Recovery Planning

---

1: **Inputs:** ($\mathcal{M}$: Markov Decision Process), ($E$: Episode Count), ($\pi_{\text{backup}}$: Task-oblivious backup Policy)
2: **Output:** ($\hat{\pi}_{\text{dmps}}$: Optimal Learned Policy)
3: $\hat{\pi}_{\text{dmps}} := \texttt{initNeuralPolicy}()$  # Initialize a neural network to act as the learned policy.
4: $\mathcal{B} := \emptyset$  # Initialize an empty replay buffer.
5: **for** $e \in [1, E]$ **do**
6:     $s := \texttt{beginEpisode}(e)$  # Initialize a new episode and receive the first observed state.
7:     **while** $\neg\texttt{terminated}(e)$ **do**
8:         $a_{\text{next}} := \hat{\pi}_{\text{dmps}}(s)$  # Choose a candidate for the next action using the learned policy.
9:         **if** $\neg\texttt{isRec}(\mathcal{T}(s, a_{\text{next}}), \pi_{\text{backup}})$ **then**
10:             $\mathcal{B} := \mathcal{B} \cup \langle s, a_{\text{next}}, \varnothing, r^- \rangle$  # Record a large negative penalty for triggering the backup policy.
11:             **if** $a_{0:n}^* = \texttt{planRec}(s, Q_{\hat{\pi}_{\text{dmps}}})$ **then** $a_{\text{next}} := a_0^*$  # Plan recovery and choose the next action.
12:             **else** $a_{\text{next}} := \pi_{\text{backup}}(s)$  # Use the task-oblivious backup policy if planning fails.
13:         $s', r := \texttt{execute}(s, a_{\text{next}})$  # Execute the selected next action on the environment.
14:         $\mathcal{B} := \mathcal{B} \cup \langle s, a_{\text{next}}, s', r \rangle$  # Update the buffer with the recent transition record.
15:         $s := s'$
16:     $\texttt{updateNeuralPolicy}(\hat{\pi}_{\text{dmps}}, \mathcal{B})$  # Update the learned policy using buffered records.
17: **return** $\hat{\pi}_{\text{dmps}}$

---

Q-function. By executing the first action on the green path and repeating dynamic recovery, the agent can demonstrate the desired behavior shown in Figure 2 (c).

Given the plan returned by $\texttt{planRec}$, the dynamic backup policy $\pi_{\text{backup}}^*$ returns the first action in the plan $a_0^*$ as its output. However, $\texttt{planRec}$ could, in theory, fail to find a safe and optimal plan, even though one exists. In such cases, $\texttt{planRec}$ returns a special symbol $\perp$, and $\pi_{\text{backup}}^*$ reverts to the task-oblivious backup policy $\pi_{\text{backup}}$. Thus, we have:

$$\pi_{\text{backup}}^*(s) = \begin{cases} a_0^* & \text{If } \texttt{planRec}(s, Q_{\hat{\pi}_{\text{dmps}}}) = a_{0:n}^* \\ \pi_{\text{backup}}(s) & \text{If } \texttt{planRec}(s, Q_{\hat{\pi}_{\text{dmps}}}) = \perp \end{cases} \tag{3}$$

An outline of $\pi_{\text{dmps}}$ is shown in Figure 1 where the control delegation mechanism is represented by the red dashed arrows (marked as ❸ and ❹) instead of the blue solid arrow (marked as ❷).

## 5.1 Planning Optimal Recovery

The specific choice of the planning algorithm to solve Equation 2 depends on the application domain. However, DMPS requires two main properties to be satisfied by the planner: *probabilistic completeness* and *asymptotic optimality*. The former property states that the planner will eventually find a solution if one exists, while the latter states that the found plan converges to the optimal solution as the allocated resources increase. There exist several state-of-the-art planners that satisfy both of these requirements, including sampling-based planners such as RRT* [78] and MCTS [79], which have been shown to be particularly effective at finding high-quality solutions in high-dimensional continuous search spaces [53, 54, 55]. These methods construct probabilistic roadmaps or search trees and deal with the exponential growth in the search space by incrementally exploring and expanding only the most promising nodes based on heuristics or random sampling. Given an implementation of $\texttt{planRec}$ that satisfies the aforementioned requirements, a significant outcome in DMPS is that, as the depth of the planning horizon ($n$) increases, the expected return from $\pi_{\text{backup}}^*$ approaches the globally optimal value. The following theorem states the optimality of recovery in $\pi_{\text{dmps}}$, in terms of exponentially diminishing recovery regret of $\pi_{\text{backup}}^*$ as $n$ increases.

**Theorem 5.1.** [2] *(Simplified)* *Suppose the use of a probabilistically complete and asymptotically optimal planner with planning horizon $n$ and sampling limit $m$. Under mild assumptions of the MDP, the recovery regret of policy $\pi_{\text{backup}}^*$ used in $\pi_{\text{dmps}}$ is almost surely bounded by order $\gamma^n$ as $m$ goes to infinity. In other words, with probability 1,*

$$\lim_{m \to \infty} RR(\pi_{\text{backup}}^*, \pi_{\text{dmps}}) = \mathcal{O}(\gamma^n).$$

## 5.2 Training Algorithm

---

[2] Extended theorem statement and proof are provided in Appendix A.1.

While our proposed recovery can be used during deployment irrespective of how the neural policy is trained, our method integrates the planner into the training loop, allowing the neural policy to learn to "imitate" the safe actions of the planner while making task-specific progress. Hence, the training loop converges to a neural policy that is both high-performant *and* safe. This is very desirable because DMPS can avoid expensive shield interventions that require on-line planning during deployment.

Algorithm 1 presents a generic deep RL algorithm for training a policy $\hat{\pi}_{\mathrm{dmps}}$. Lines 3-4 of the algorithm perform initialization of the neural policy $\hat{\pi}_{\mathrm{dmps}}$ as well as the *replay buffer* $\mathcal{B}$, which stores a set of tuples $\langle s, a, s', r \rangle$ that capture transitions and their corresponding reward $r$. Each training episode begins with the agent observing the initial state $s$ (line 6) and terminates when the goal state is reached or after a predetermined number of steps are taken (line 7). The agent first attempts to choose its next action $a_{\mathrm{next}}$ using the learned policy $\hat{\pi}_{\mathrm{dmps}}$ (Line 8). If the execution of $a_{\mathrm{next}}$ leads to a non-recoverable state according to $\pi_{\mathrm{backup}}$ (line 9), the algorithm first adds a record to the replay buffer where the current state and action are associated with a high negative reward $r^-$ (line 10). It then performs dynamic model predictive shielding to ensure safety (lines 11-12) as discussed earlier: If the planner yields a valid policy, the the next action is chosen as the first action in the plan (line 11); otherwise, the backup policy $\pi_{\mathrm{backup}}$ is used to determing the next action. Then, $a_{\mathrm{next}}$ is executed at line 13 to obtain a new state $s'$ and its corresponding reward $r$. This new transition and its corresponding reward are again added to the replay buffer (line 14), which is then used to update the neural policy at line 16, after the termination of the current training episode.

## 6 Experiments

In this section, we present an empirical study of DMPS on 13 benchmarks and compare it against 4 baselines. The details of our implementation and experimental setup are presented in Appendix A.3.

***Benchmarks.*** We evaluate our approach on five *static benchmarks* (ST), where the agent's environment is static, and eight *dynamic benchmarks*, where the agent's environment includes moving objects. Static benchmarks (obstacle, obstacle2, mount-car, road, and road2d) are drawn from prior work [16, 27] and include classic control problems. The more challenging dynamic benchmarks (dynamic-obst, single-gate, double-gates, and double-gates+) require the agent to adapt its policy to accommodate complex obstacle movements. Specifically, dynamic-obs features moving obstacles along the agent's path. In single-gate, a rotating circular wall with a small opening surrounds the goal position. double-gate is similar but includes *two* concentric circular walls around the goal, and double-gate+ is the most challenging, featuring increased wall thickness to force more efficient navigation through the openings. For each dynamic benchmark, we consider two different agent dynamics: *differential drive dynamics* (DD), featuring an agent with two independently driven wheels, and *double integrator dynamics* (DI), where the agent's acceleration in any direction can be adjusted by the policy. A detailed description of benchmarks can be found in Appendix A.3.

***Baselines.*** We compare DMPS (with a planning horizon of $n = 5$) with five baselines. Our first baseline is MPS, the standard model predictive shielding approach. The second baseline is REVEL [16], a recent PSRL approach that learns verified neurosymbolic policies through iterative mirror descent. REVEL requires a white-box model of the environment's worst-case dynamics, which is challenging to develop for dynamic benchmarks; hence, we apply REVEL only to static benchmarks. We also compare DMPS against three established RL methods: CPO [3], PPO-Lag (PPO Lagrangian) [13, 14], and TD3 [15]. All aim to reduce safety violations during training by incorporating safety constraints into the objective. In TD3, a negative penalty is applied to unsafe steps. In CPO, a fixed number of violations are tolerated. Finally, in PPO-Lag, a Lagrangian multiplier is used to balance safety cost with reward.

### 6.1 Safety Results

Table 1 presents our experimental results regarding safety. All results are averaged over 5 random seeds. For PSRL methods (DMPS, MPS, and REVEL), which guarantee worst-case safety, we report the average number of shield invocations per episode. Generally, less frequent shield invocation indicates higher performance of the approach. For SRL methods (TD3, PPO-Lag and CPO), we report the average number of safety violations per episode.

Table 1: Safety Results

| Benchmark | | # Shield Invocations / Episode | | | | | | # Safety Violations / Episode | | | | | |
|---|---|---|---|---|---|---|---|---|---|---|---|---|---|
| | | DMPS | | MPS | | REVEL | | CPO | | PPO-lag | | TD3 | |
| | | mean | sd | mean | sd | mean | sd | mean | sd | mean | sd | mean | sd |
| ST | obstacle | **0.0** | 0.0 | **0.0** | 0.0 | 4.0 | 8.0 | **0.0** | 0.0 | **0.0** | 0.0 | **0.0** | 0.0 |
| | obstacle2 | **0.0** | 0.0 | **0.0** | 0.0 | 33.8 | 30.3 | 0.9 | 0.2 | 6.42 | 0.19 | **0.0** | 0.0 |
| | mount-car | **0.0** | 0.0 | **0.0** | 0.0 | 4.0 | 4.0 | 2.0 | 2.1 | 22.2 | 28.9 | **0.0** | 0.0 |
| | road | **0.0** | 0.0 | **0.0** | 0.0 | 0.8 | 0.74 | **0.0** | 0.0 | **0.0** | 0.0 | **0.0** | 0.0 |
| | road2d | **0.0** | 0.0 | **0.0** | 0.0 | 0.0 | 0.0 | **0.0** | 0.0 | **0.0** | 0.0 | **0.0** | 0.0 |
| DI | dynamic-obst | **9.3** | 1.8 | 144.1 | 26.2 | - | - | 1.7 | 0.8 | 3.6 | 3.9 | **0.8** | 1.2 |
| | single-gate | **0.1** | 0.0 | 0.2 | 0.1 | - | - | 2.0 | 1.2 | 10.0 | 5.5 | **0.0** | 0.0 |
| | double-gates | **4.5** | 1.2 | 28.3 | 6.9 | - | - | 2.1 | 1.7 | 11.8 | 6.3 | **0.3** | 0.5 |
| | double-gates+ | **24.2** | 6.4 | 239.8 | 16.3 | - | - | 2.9 | 1.0 | 6.5 | 4.5 | **0.0** | 0.0 |
| DD | dynamic-obst | **105.2** | 39.9 | 144.9 | 39.9 | - | - | 1.7 | 1.9 | 2.9 | 2.2 | **0.8** | 0.16 |
| | single-gate | **3.1** | 1.6 | 7.4 | 6.9 | - | - | 5.2 | 3.5 | 6.7 | 7.2 | **0.1** | 0.2 |
| | double-gates | **5.5** | 1.9 | 52.5 | 17.6 | - | - | 3.9 | 1.7 | 7.9 | 9.2 | **3.0** | 5.5 |
| | double-gates+ | **18.4** | 13.0 | 106.2 | 18.5 | - | - | 12.1 | 2.9 | 6.9 | 6.1 | **0.2** | 0.4 |

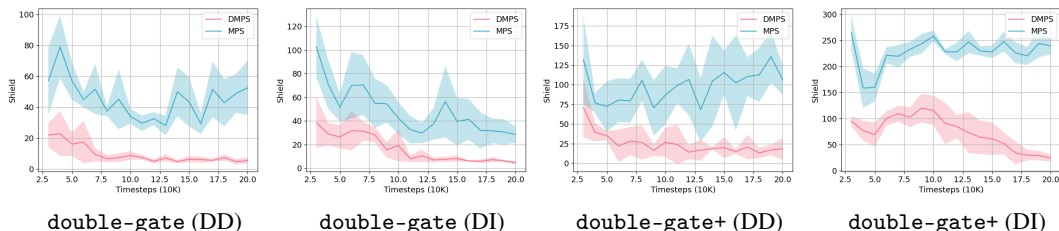

| double-gate (DD) | double-gate (DI) | double-gate+ (DD) | double-gate+ (DI) |
|---|---|---|---|

Figure 3: Shield Invocations in `double-gate` and `double-gate+`

Due to the relative simplicity of the static benchmarks, the PSRL baselines achieve safety with very infrequent shield invocations. Even the SRL baselines are mostly able to avoid safety violations in these benchmarks. On the other hand, in dynamic benchmarks, PSRL approaches heavily rely on shielding to achieve safety, and SRL approaches violate safety more frequently. Notably, the number of DMPS shield invocations is significantly lower than in other baselines. Across all dynamic benchmarks, DMPS invokes the shield an average of 24.7 times per episode, whereas MPS triggers the shield 124.1 times. The results also indicate that the standard deviation values for shield invocations in DMPS are consistently lower than those of MPS, indicating more stable and predictable performance.

Figure 3 shows the number of shield triggers plotted against episodes for the `double-gate` and `double-gate+` dynamic benchmarks. The error regions are 1-sigma over random seeds. Under both agent dynamics, DMPS achieves significantly fewer shield invocations compared to MPS. Moreover, the number of shield invocations in DMPS consistently decreases as training progresses, whereas this trend is not present in MPS. In fact, in many scenarios, the number of shield invocations in MPS increases with more training because the learned policy reduces its exploratory actions and adheres more strictly to a direct path to the goal. However, this frequently leads to being blocked by obstacles and results in repeated shield invocations.

## 6.2 Performance Results

Table 2 presents the per-episode return over the last 10 test episodes of a run. The reported mean and standard deviations are computed over 5 random seeds. The results indicate that DMPS and MPS exhibit comparable performance across most static benchmarks, with the exception of the more challenging `obstacle` and `obstacle2` benchmarks for which DMPS significantly outperforms MPS. In dynamic benchmarks, DMPS outperforms MPS in all benchmarks except for `single-gate` (DI), where both methods achieve equivalent results. Both DMPS and MPS consistently outperform REVEL. The SRL approaches (CPO, PPO-Lag, and TD3) perform reasonably well in most static benchmarks but their performance significantly deteriorates in dynamic benchmarks, failing to achieve positive returns in any instance. This is because the policies in CPO, PPO-Lag, and TD3 rapidly learn to avoid both the obstacles and the goal due to harsh penalties for safety violations, thus accruing the negative rewards for each timestep spent away from the goal.

Table 2: Performance Results

| Benchmark | | Total Return in Final 10 Episode | | | | | | | | | | |
|---|---|---|---|---|---|---|---|---|---|---|---|---|
| | | DMPS | | MPS | | REVEL | | CPO | | PPO-Lag | | TD3 | |
| | | mean | sd | mean | sd | mean | sd | mean | sd | mean | sd | mean | sd |
| ST | obstacle | 32.7 | 0.3 | 8.6 | 47.9 | −41.6 | 52.7 | 32.8 | 0.0 | 32.8 | 0.4 | **32.9** | 0.0 |
| | obstacle2 | 20.2 | 15.2 | −1.8 | 3.2 | 9.3 | 21.1 | 33.0 | 0.2 | **34.1** | 0.1 | −1.2 | 3.0 |
| | mount-car | 81.2 | 0.3 | **85.1** | 1.8 | 11.4 | 34.1 | 9.6 | 35.3 | −21.3 | 2.9 | −30.0 | 35.5 |
| | road | **22.7** | 0.0 | **22.7** | 0.0 | 9.7 | 16.4 | **22.7** | 0.0 | **22.7** | 0.05 | **22.7** | 0.0 |
| | road2d | 24.0 | 0.2 | 24.0 | 0.2 | 11.2 | 16.5 | 24.0 | 0.0 | 24.0 | 0.1 | 24.0 | 0.1 |
| DI | dynamic-obs | **13.2** | 0.0 | −1.3 | 1.9 | - | - | −21.4 | 21.1 | −4.2 | 24.9 | −5.0 | 0.3 |
| | single-gate | **11.6** | 0.0 | **11.6** | 0.0 | - | - | −19.6 | 17.0 | −2.0 | 1.1 | −2.1 | 0.2 |
| | double-gates | **12.7** | 1.0 | 11.5 | 1.1 | - | - | −6.7 | 5.4 | −3.9 | 11.1 | −3.6 | 1.0 |
| | double-gates+ | **13.0** | 0.6 | −0.9 | 0.1 | - | - | −17.4 | 12.8 | −2.8 | 0.4 | −4.1 | 1.1 |
| DD | dynamic-obst | **7.4** | 3.7 | 6.3 | 2.1 | - | - | −4.5 | 0.5 | −3.6 | 0.8 | −5.3 | 0.5 |
| | single-gate | **11.5** | 0.1 | 11.4 | 0.1 | - | - | −2.5 | 0.3 | −2.4 | 0.2 | −3.1 | 0.5 |
| | double-gates | **13.1** | 0.5 | 10.9 | 1.8 | - | - | −2.4 | 0.6 | −1.9 | 0.9 | −3.4 | 0.5 |
| | double-gates+ | **13.0** | 0.6 | 8.5 | 2.3 | - | - | −2.5 | 0.3 | −20.7 | 22.8 | −3.6 | 0.3 |

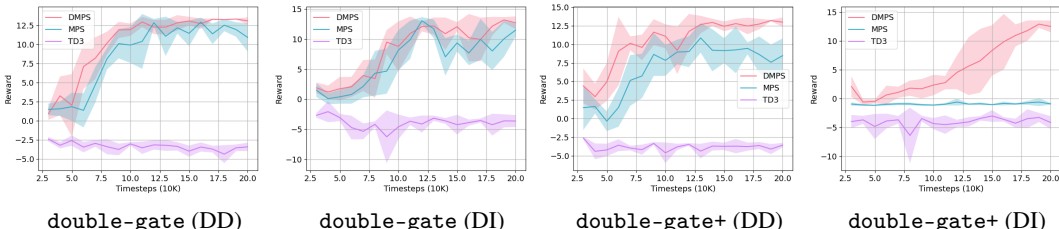

double-gate (DD)     double-gate (DI)     double-gate+ (DD)     double-gate+ (DI)

Figure 4: Episodic Returns in `double-gate` and `double-gate+`

Figure 4 presents the total return plotted against the episode number for the `double-gate` and `double-gate+` dynamic benchmarks. The error regions are 1-sigma over random seeds. The `CPO` and `PPO-Lag` baselines are omitted due to their significantly poor performance, which distorts the scale of the graphs. In all benchmarks, `DMPS` demonstrates superior performance compared to the other methods. While `MPS` performs adequately in three of the benchmarks, it exhibits poor performance in the `double-gate+` (DI) benchmark.

### 6.3 Analysis

We analyze the performance of `DMPS` and `MPS` on the `double-gate+` environment under double-integrator dynamics. Figure 5a shows trajectories from the first half of training when the agent's policy and $Q$ function are still under-trained. When the agents approach the first gate and attempt to cross it, the shield is triggered. In the case of `MPS`, this shield simply pushes the agent back outside the gate (see blue trajectory), and the agent is unable to make any progress. In contrast, `DMPS` plans actions that maximize an $n$-step return target, allowing the agent to initially make progress. However, we

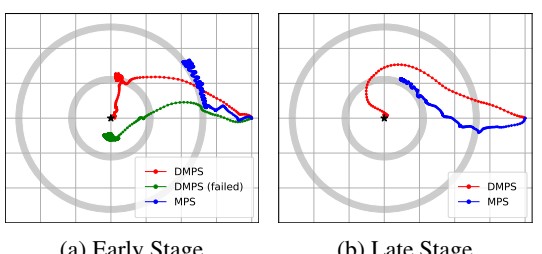

(a) Early Stage     (b) Late Stage

Figure 5: Example trajectories in `double-gate+` .

can observe from the green trajectory that the agent's $Q$ function and policy are under-trained: after having made it through the obstacles, the agent is not sufficiently goal-aware to reliably make it to the center. This makes the `DMPS` planner objective inaccurate with respect to long-term return, causing suboptimal actions to be selected by the shield. In the red trajectory, for instance, the agent spends a large portion of the trajectory trying to cross the second gate, only to retreat and get stuck in that position until eventually getting through.

As training progresses, the `MPS` backup policy hinders the agent's exploration of the environment, impeding the learning of the neural policy. By the end, most runs of the `MPS` agent are unable to progress past the first gate. Figure 5b shows one of the few `MPS` runs that successfully make it through the first gate towards the end of training; however, the `MPS` agent still fails to make it through

the second gate. In contrast, the `DMPS` agent can learn from the actions suggested by the planner, improving its policy and $Q$ function. This improvement makes the `DMPS` planner's objective a more accurate estimation of long-term return, further strengthening the planner. Consequently, the `DMPS` agent demonstrates significantly better behaviors, as shown by the red trajectory in Figure 5b.

## 7 Limitations

### 7.1 Determinism

First, our approach requires a perfect-information deterministic model, which could limit its usability in real-world deployment. Much prior work on provably safe locomotion makes the same determinism assumptions that we do [10, 31, 80], with some such algorithms even having been deployed on real robots [3]. However, extending our DMPS approach to stochastic environments is a promising direction for future work. In particular, since prior MPS work has been extended to work in stochastic settings [40], we believe our DMPS approach can be similarly extended to the stochastic setting.

### 7.2 Computational Overhead

Another potential limitation of our approach is the computational overhead of the planner. We use MCTS in an anytime planning setting, where we devote a fixed number of node expansions to searching for a plan. The clock time needed is linear in the allocated compute budget. However, the worst-case computational complexity to densely explore the search space, as in general planning problems, would be $O(\exp(H))$ since the planning search space grows exponentially. Our implementation used MCTS with 100 node expansions, a plan horizon of 5, and a branch factor of 10, which amounts to exploring 1000 states in total. On average, we found that when the shield is triggered, planning takes 0.4 seconds per timestep. The code we used is unoptimized, written in Python, and single-threaded. Since MCTS is a CPU-intensive search process, switching to a language C++ would likely yield significant speed improvements, and distributing computation over multiple cores would further slash the compute time by the number of assigned cores. We perform some additional experiments on the `double-gates+` environment, outlined in subsection A.4, to see how necessary compute budget scales with planner depth, confirming a rough exponential relationship.

### 7.3 Sufficiency of Planning Horizon

Finally, it can be asked whether small planning horizons (in our case, we used $H = 5$) are sufficient to solve tricky planning problems. Our planner objective is designed to ensure that the planner accounts for both short-term and long-term objectives, preventing overly myopic behavior even with short horizons. However, the length of the horizon still affects how close to the globally optimal solution the result is, with a tradeoff of computational cost. To see this empirically, we conducted an experiment on the `double-gates+` environment, re-evaluating it under different planner depths and observing performance differences. Despite better initial performance, the low horizon agent converged to the same performance once the policy had fully stabilized. As part of our analysis, we also found that trivial planning ($H = 1$) does not work, reaffirming the necessity of a planner. More analysis and experimental results can be found in subsection A.5.

## 8 Conclusions

In this paper, we proposed Dynamic Model Predictive Shielding (`DMPS`), which is a variant of Model Predictive Shielding (MPS) that performs dynamic recovery by leveraging a local planner. The proposed approach takes less conservative actions compared to MPS while still guaranteeing safety, and it learns neural policies that are both effective and safe in practice. Our evaluation on several challenging tasks shows that `DMPS` improves upon the state-of-the-art methods in Safe Reinforcement Learning.

***Impact Statement.*** Over the past decade, reinforcement learning has experienced significant advancements and is increasingly finding applications in critical safety environments, such as autonomous driving. The stakes in such settings are high, with potential failures risking considerable property damage or loss of life. This study seeks to take a meaningful step for mitigating these risks by developing reinforcement learning agents that are rigorously aligned with real-world safety requirements.

***Acknowledgements.*** This work is partially supported by the National Science Foundation (CCF-2319471 and CCF-1901376). Any opinions, findings, and conclusions expressed in this material are those of the authors and do not necessarily reflect the views of the sponsors.

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

# A  Supplemental Materials

In this section, we provide supplemental material complementary to the main content of the paper. This includes a complete proof for Theorem 5.1 and additional details omitted from section 6 due to space constraints.

## A.1  Proof of Theorem 5.1

Throughout this section, we denote $Q, Q^\star$, and $V^\star$ to be the agent $Q$ function, the optimal $Q$ function, and the optimal $V$ function respectively.

Our planner is parameterized by a fixed plan depth $n$ and a sampling limit $m$. The planner is tasked with finding actions $a_{0:n}$ to optimize the objective $J_Q(s_0, a_{0:n}) = \left(\sum_{i=0}^{n-1} \gamma^i R_i\right) + \gamma^n Q(s_n, a_n)$. Let $\epsilon_m$ be the supremum of $J_Q(s, a_{0:n}^\star) - J_Q(s, a_{0:n})$ over $s \in \mathcal{S}$, where $a_{0:n} = \texttt{planRec}(s)$ and $a_{0:n}^\star$ is the optimal sequence of actions with respect to the $J_Q$ objective.

We assume that our planner is probabilistically complete and asymptotically optimal. In context, this means that $\lim_{m\to\infty} \epsilon_m = 0$ with probability 1. With this grounding established, we state the extended version of Theorem 5.1.

**Theorem A.1.** *With probability 1, we have:*

1. *Under the assumption that $Q$ is $\epsilon-$close to $Q^\star$, we have $\lim_{m\to\infty} RR(\pi_{backup}^\star, \pi_{dmps}) = \mathcal{O}(\epsilon\gamma^n)$.*

2. *Under the assumptions that the maximum reward per timestep is bounded, that $\mathcal{S} \times \mathcal{A}$ is compact, and that $Q$ is a continuous function, we have $\lim_{m\to\infty} RR(\pi_{backup}^\star, \pi_{dmps}) = \mathcal{O}(\gamma^n)$.*

3. *Under the assumptions that $Q$ and $Q^\star$ are both Lipschitz continuous, that $\|\mathcal{T}(s,a) - s\|_2$ is bounded over all $(s,a) \in \mathcal{S} \times \mathcal{A}$, and that the initial state space $\mathcal{S}_0$ and the action space $\mathcal{A}$ are both bounded, we have $\lim_{m\to\infty} RR(\pi_{backup}^\star, \pi_{dmps}) = \mathcal{O}(n\gamma^n)$.*

We first show the following general lemma.

**Lemma A.2.** *Fix a state $s \in \mathcal{S}$, and let $a = \pi_{backup}^\star(s)$ be the first action returned by* planRec. *Let $\mathcal{S}_n$ be the set of all states reachable from $s$ in $n$ steps, and let $B = \sup_{s' \in \mathcal{S}_n, a' \in \mathcal{A}} |Q^\star(s', a') - Q(s', a')|$. Then, $0 \leq V^\star(s) - Q^\star(s,a) \leq 2B\gamma^n + \epsilon_m$.*

*Proof.* The bound $V^\star(s) - Q^\star(s,a) \geq 0$ follows trivially from the definitions of $V^\star$ and $Q^\star$. We now prove the upper bound.

Denote $a_{0:n}$ as the sequence of actions returned by the planner. Note that $a = a_0$. Denote $a_{0:n}^\star$ as the sequence of actions that are optimal with respect to true infinite-horizon return from $s$ (not necessarily with respect to $J_Q$).

We allow $R_{0:n-1}$ and $R_{0:n-1}^\star$ to be the reward sequences attained by rolling out $a_{0:n-1}$ and $a_{0:n-1}^\star$ respectively. Similarly, we allow $s_n$ and $s_n^\star$ to be the states reached at the end of executing $a_{0:n-1}, a_{0:n-1}'$, and $a_{0:n-1}^\star$ respectively.

First, we have

$$Q^\star(s,a) \geq \left(\sum_{i=0}^{n-1} \gamma^i R_i\right) + \gamma^n Q^\star(s_n, a_n).$$

This follows immediately from the definition of $Q^\star$ and the fact that $a = a_0$.

Since $|Q^\star(s_n, a_n) - Q(s_n, a_n)| \leq B$, we have

$$\left(\sum_{i=0}^{n-1} \gamma^i R_i\right) + \gamma^n Q^\star(s_n, a_n) \geq \left(\sum_{i=0}^{n-1} \gamma^i R_i\right) + \gamma^n Q(s_n, a_n) - B\gamma^n$$

$$= J_Q(s, a_{0:n}) - B\gamma^n.$$

We know that $a_{0:n}$ optimizes $J_Q$ to within tolerance of $\epsilon_m$, so we can write $J_Q(s, a_{0:n}) \geq J_Q(s, a_{0:n}^\star) - \epsilon_m$. We get

$$J_Q(s, a_{0:n}) - B\gamma^n \geq J_Q(s, a_{0:n}^\star) - \epsilon_m - B\gamma^n$$
$$= \left(\sum_{i=0}^{n-1} \gamma^i R_i^\star\right) + \gamma^n Q(s_n^\star, a_n^\star) - \epsilon_m - B\gamma^n$$

Invoking again $|Q^\star(s_n^\star, a_n^\star) - Q(s_n^\star, a_n^\star)| \leq B$, we get

$$\left(\sum_{i=0}^{n-1} \gamma^i R_i^\star\right) + \gamma^n Q(s_n^\star, a_n^\star) - \epsilon_m - B\gamma^n \geq \left(\sum_{i=0}^{n-1} \gamma^i R_i^\star\right) + \gamma^n Q^\star(s_n^\star, a_n^\star) - \epsilon_m - 2B\gamma^n$$
$$= V^\star(s) - \epsilon - 2B\gamma^n.$$

Putting everything together gives us $Q^\star(s, a) \geq V^\star(s) - \epsilon - 2B\gamma^n$. Rearranging this gives us the desired claim. $\qquad\square$

With this, we can begin proving the main theorem.

**Lemma A.3.** *(Part 1 of Thm A.1) Under the assumption that $Q$ is $\epsilon$-close to $Q^\star$, we have* $lim_{m\to\infty} RR(\pi_{backup}^\star, \pi_{dmps}) = \mathcal{O}(\epsilon\gamma^n)$ *with probability 1.*

*Proof.* In the context of Lemma A.1., we see that $B \leq \epsilon$. Thus, we conclude that for all states $s \in \mathcal{S}$, we have $V^\star(s) - Q^\star(s, \pi_{dmps}(s)) \leq 2\epsilon\gamma^n + \epsilon_m$. Since $RR$ is simply an expectation of $V^\star(s) - Q^\star(s, \pi_{backup}^\star(s))$ over some state distribution over $\mathcal{S}$, we can conclude $RR(\pi_{backup}^\star, \pi_{dmps}) \leq 2\epsilon\gamma^n + \epsilon_m$. We know from asymptotic optimality assumption that $\epsilon_m$ decays to 0 as $m$ goes to infinity, so we get $\lim_{m\to\infty} RR(\pi_{backup}^\star, \pi_{dmps}) = \mathcal{O}(\epsilon\gamma^n)$. $\qquad\square$

**Lemma A.4.** *(Part 2 of Thm A.1) Under the assumption that the maximum reward per timestep is bounded, that $\mathcal{S} \times \mathcal{A}$ is compact, and that $Q$ is a continuous function, we have* $\lim_{m\to\infty} RR(\pi_{backup}^\star, \pi_{dmps}) = \mathcal{O}(\gamma^n)$ *with probability 1.*

*Proof.* It suffices to show that $|Q^\star(s, a) - Q(s, a)|$ is bounded over all $(s, a) \in \mathcal{S} \times \mathcal{A}$. We can then simply apply Lemma A.3 to get the $\mathcal{O}(\gamma^n)$ bound.

It is known that a continuous function with compact domain has a maximum and minimum value. Thus, there exists some $C_Q$ such that $|Q(s, a)| < C_Q$ for all $(s, a) \in \mathcal{S} \times \mathcal{A}$.

Pick a constant $R$ such that the absolute value of the reward at any timestep is less than $R$. We see that $Q^\star(s, a)$ is less than $\sum_{t=0}^\infty R\gamma^t = \frac{R}{1-\gamma}$ and greater than $\sum_{t=0}^\infty (-R)\gamma^t = \frac{-R}{1-\gamma}$. Consequently, letting $C_{Q^\star} = \frac{R}{1-\gamma}$, we get $|Q^\star(s, a)| < C_{Q^\star}$ over all $(s, a) \in \mathcal{S} \times \mathcal{A}$.

With this, $|Q^\star(s, a) - Q(s, a)| \leq |Q^\star(s, a)| + |Q(s, a)| < C_Q + C_{Q^\star}$ over all $(s, a) \in \mathcal{S} \times \mathcal{A}$. This shows the boundedness we needed to attain the asymptotic bound. $\qquad\square$

**Lemma A.5.** *(Part 3 of Thm A.1) Under the assumptions that $Q$ and $Q^\star$ are both Lipschitz continuous, that $\|\mathcal{T}(s, a) - s\|_2$ is bounded over all $(s, a) \in \mathcal{S} \times \mathcal{A}$, and that the initial state space $\mathcal{S}_0$ and the action space $\mathcal{A}$ are both bounded, we have* $\lim_{m\to\infty} RR(\pi_{backup}^\star, \pi_{dmps}) = \mathcal{O}(n\gamma^n)$ *with probability 1.*

*Proof.* Suppose that both $Q$ and $Q^\star$ are $K$-Lipschitz continuous. Let $d_\mathcal{A}$ be the diameter of $\mathcal{A}$. Let $d_\mathcal{T}$ be the supremum of $\|\mathcal{T}(s, a) - s\|_2$ over $(s, a) \in \mathcal{S} \times \mathcal{A}$. We fix some arbitrary $s \in \mathcal{S}_0$ and $a \in \mathcal{A}$. Intuitively, this will act as a "central" point upon which we can estimate values for other states. Let $D = |Q(s, a) - Q^\star(s, a)|$. Let $d_0$ be the radius of $\mathcal{S}_0$ when centered at $s$.

We first prove a subclaim. Take an arbitrary $s' \in \mathcal{S}$, and let $d$ be the $\ell_2$ distance between $s$ and $s'$. Then,

$$V^\star(s') - Q^\star(s', \pi^\star_{\text{backup}}(s')) \le 2\gamma^n \left( D + 2K\sqrt{d_{\mathcal{A}}^2 + (d + nd_{\mathcal{T}})^2} \right) + \epsilon_m.$$

The idea is to invoke Lemma A.1. To do so, we need to attain a bound on $B$ as defined by the Lemma. Consider a state $s''$ that is $n$ steps away from $s'$. The Triangle Inequality lets us bound the distance between $s'$ and $s''$ to $d_{\mathcal{T}}$. A second application of the Triangle Inequality lets us bound the distance between $s$ and $s''$ to $d + d_{\mathcal{T}}$. We can further use the boundedness of the action space to note that the maximum distance between the state-action pair $(s, a)$ and $(s'', a'')$ for an arbitrary $a'' \in \mathcal{A}$ is $d'' = \sqrt{d_{\mathcal{A}}^2 + (d + nd_{\mathcal{T}})^2}$. Finally, we use Lischpitz continuity to get $|Q^\star(s, a) - Q^\star(s'', a'')| \le Kd''$ and $|Q(s, a) - Q(s'', a'')| \le Kd''$. With this,

$$|Q^\star(s'', a'') - Q(s'', a'')|$$
$$\le |Q^\star(s'', a'') - Q^\star(s, a)| + |Q(s'', a'') - Q(s, a)| + |Q^\star(s, a) - Q(s, a)|$$
$$\le 2Kd'' + D.$$

Using this as a bound on $B$ and invoking Lemma A.1. demonstrates the subclaim.

Now, consider some arbitrary trajectory $s_0, a_0, s_1, a_2, \ldots, s_{k-1}, a_{k-1}, s_k$, with $s_0 \in \mathcal{S}_0$ and some fixed non-negative integer $k$. Via repeated application of the Triangle Inequality, one can bound the distance between $s$ and $s_k$ to at most $d_0 + kd_{\mathcal{T}}$. Invoking the subclaim, we see that

$$V^\star(s_k) - Q^\star(s_k, \pi^\star_{\text{backup}}(s_k)) \le 2\gamma^n \left( D + 2K\sqrt{d_{\mathcal{A}}^2 + (d_0 + kd_{\mathcal{T}} + nd_{\mathcal{T}})^2} \right) + \epsilon_m.$$

We can repeatedly use the fact that $k + n \ge 1$ to clean up the bound here. Namely, we write

$$D + 2K\sqrt{d_{\mathcal{A}}^2 + (d_0 + kd_{\mathcal{T}} + nd_{\mathcal{T}})^2}$$
$$\le D(k + n) + 2K\sqrt{(k + n)^2 d_{\mathcal{A}}^2 + ((k + n)d_0 + kd_{\mathcal{T}} + nd_{\mathcal{T}})^2}$$
$$= (k + n)\left[ D + 2K\sqrt{d_{\mathcal{A}}^2 + (d_0 + d_{\mathcal{T}})^2} \right].$$

Setting $C = D + 2K\sqrt{d_{\mathcal{A}}^2 + (d_0 + d_{\mathcal{T}})^2}$, we can sum up the equations above by writing

$$V^\star(s_k) - Q^\star(s_k, \pi^\star_{\text{backup}}(s_k)) \le 2C\gamma^n(k + n) + \epsilon_m.$$

Now, we can finally bound $RR$. Let $\rho^{\pi_{\text{dmps}}}$ be the discounted state visitation distribution. We define $\rho_t^{\pi_{\text{dmps}}}$ to be the state visitation distribution at timestep $t$. We get

$$RR(\pi^\star_{\text{backup}}, \pi_{\text{dmps}}) = \mathbb{E}_{s \sim \rho^{\pi_{\text{dmps}}}}[\mathbb{I}_{backup} \cdot (V^\star(s) - Q^\star(s, \pi^\star_{\text{backup}}(s)))]$$
$$\le \mathbb{E}_{s \sim \rho^{\pi_{\text{dmps}}}}[V^\star(s) - Q^\star(s, \pi^\star_{\text{backup}}(s))]$$
$$= \int_{\mathcal{S}} [V^\star(s) - Q^\star(s, \pi^\star_{\text{backup}}(s))]\rho^{\pi_{\text{dmps}}}(s)ds$$
$$= \int_{\mathcal{S}} [V^\star(s) - Q^\star(s, \pi^\star_{\text{backup}}(s))]\left[ (1 - \gamma)\sum_{k=0}^{\infty} \gamma^k \rho_k^{\pi_{\text{dmps}}}(s) \right] ds$$
$$= \sum_{k=0}^{\infty} (1 - \gamma)\gamma^k \int_{\mathcal{S}} [V^\star(s) - Q^\star(s, \pi^\star_{\text{backup}})]\rho_k^{\pi_{\text{dmps}}}(s)ds$$
$$= \sum_{k=0}^{\infty} (1 - \gamma)\gamma^k \mathbb{E}_{s \sim \rho_k^{\pi_{\text{dmps}}}}[V^\star(s) - Q^\star(s, \pi^\star_{\text{backup}}(s))]$$

---

**Algorithm 2** Monte Carlo Tree Search (`planRec`)

---

1: **Inputs:** ($s_1$: Start State), ($Q$: state-action function)
2: **Hyper-parameters:** ($K$: Branching Factor), ($I$: Iteration Count), ($L$: Maximum Path Length)
3: $\hat{Q} := \texttt{initHashmap}(\mathcal{S} \times \mathcal{A} \to \mathbb{R})$  # Initialize empty data structure for state-action value estimates.
4: $N := \texttt{initHashmap}(\mathcal{S} \times \mathcal{A} \to \mathbb{N})$  # Initialize empty data structure for state-action visit counts.
5: $T := \texttt{initTree}(s_1)$  # Initialize a tree with start state $s_0$ as the root.
6: $res := \texttt{expand}(s_1, T, \hat{Q}, N)$  # Attempt expanding the tree root with actions leading to recoverable states.
7: **if** $\neg res$ **then return** $\bot$  # Return $\bot$ if the root state cannot be expanded.
8: **for** $t \in [1, I]$ **do**
9:     # Select a path of length less than $L$ from the root using the Upper Confidence Bound (UCB) formula.
10:     $(s_1, a_1, R_1), \ldots, (s_{r-1}, a_{r-1}, R_{r-1}), s_r := \texttt{selectPathUCB}(\hat{Q}, L)$
11:     **if** $\texttt{isLeaf}(s_r, T)$ **then**
12:         $\texttt{expand}(s_r, T, \hat{Q}, Q, N)$  # Expand the leaf state at the end of the path with actions leading to recoverable states.
13:     $a_r := \texttt{selectAction}(s_r, \hat{Q})$  # Choose the best outgoing action from $s_r$ with respect to $\hat{Q}$.
14:     **for** $i \in [2, r]$ **do**
15:         # Update the state-action estimates with rewards collected on the selected path.
16:         $\hat{Q}(s_i, a_i) := \frac{1}{N(s_i, a_i)+1} \left[ N(s_i, a_i) \cdot \hat{Q}(s_i, a_i) + \left( \left( \sum_{j=i}^{r-1} \gamma^{j-i} \cdot R_j \right) + \gamma^{r-i} \cdot \hat{Q}(s_r, a_r) \right) \right]$
17:     **for** $i \in [2, r]$ **do**
18:         $N(s_i, a_i) := N(s_i, a_i) + 1$  # Increment state-action visit count for the selected path.
19: $(s_1, a_1), (s_2, a_2), \ldots, (s_n, a_n) := \texttt{selectPathGreedy}(\hat{Q})$  # Use $\hat{Q}$ to greedily select the optimal path from root to a leaf.
20: **return** $(a_1, \ldots, a_n)$  # Return the actions on the greedily selected path.

---

**Algorithm 3** Sampling-Based Tree Expansion (`expand`)

---

1: **Inputs:** ($s$: Expanding State), ($T$: State-Action Tree), ($\hat{Q}$: Local Value Estimate), ($Q$: Long-term Value Function), ($N$: Visit Count)
2: $a_1, \ldots, a_K \sim \mathcal{A}$  # Sample $K$ actions.
3: $success := False$
4: **for** $i \in [1, K]$ **do**
5:     **if** $\mathcal{T}(s, a_i) \in \mathcal{S}_{rec}$ **then**
6:         $\hat{Q}(s, a_i) := Q(s, a_i)$  # Initialize value estimate for $(s, a_i)$.
7:         $N(s, a_i) := 1$  # Record first visit for $(s, a_i)$ pair.
8:         $\texttt{updateTree}(T, s, a_i)$  # Add $(s, a_i)$ to the tree.
9:         $success := True$  # Record at least one recoverable action was found.
10: **return** $success$

---

$$\leq \sum_{k=0}^{\infty} (1-\gamma)\gamma^k [2C\gamma^n(k+n) + \epsilon_m]$$

$$= \left[ 2C(1-\gamma)\gamma^n \sum_{k=0}^{\infty} \gamma^k(k+n) \right] + \sum_{k=0}^{\infty}(1-\gamma)\gamma^k \epsilon_m$$

$$= 2C(1-\gamma)\gamma^n \left[ \left( \sum_{k=0}^{\infty} k\gamma^k \right) + n \sum_{k=0}^{\infty} \gamma^k \right] + \epsilon_m.$$

We can evaluate the arithmetic-geometric series $\sum_{k=0}^{\infty} k\gamma^k$ to equal $\frac{\gamma}{(1-\gamma)^2}$. The geometric sequence $\sum_{k=0}^{\infty} \gamma^k$ evaluates to $\frac{1}{1-\gamma}$. Thus, our sum becomes

$$2C(1-\gamma)\gamma^n \left[ \frac{\gamma}{(1-\gamma)^2} + \frac{n}{1-\gamma} \right] + \epsilon_m = \mathcal{O}(n\gamma^n) + \epsilon_m.$$

Due to asymptotic optimality, taking the limit of this as $m$ goes to infinity gives a bound of $\mathcal{O}(n\gamma^n)$. $\square$

## A.2  Monte Carlo Tree Search

The planning function `planRec` used in DMPS is realized via Monte Carlo Tree Search (MCTS) [79], which facilitates efficient online planning by balancing exploration and exploitation. MCTS has been shown to be highly effective for planning in large search spaces with either discrete [52, 81, 51] or continuous [54, 57, 82] states and actions. DMPS utilizes a continuous variant of MCTS that employs sampling-based methods to manage the exponential increase in search space size as the planning horizon expands [53]. A high-level overview of this approach follows. Algorithms 2 and 3 present our implementation of MCTS.

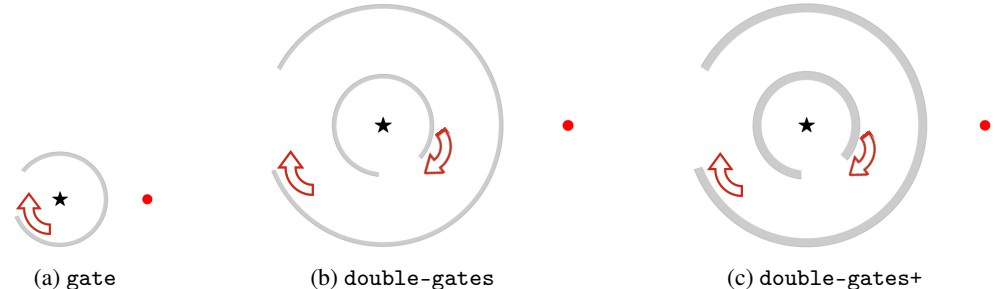

|  (a) `gate` | (b) `double-gates` | (c) `double-gates+` |

Figure 6: Visualization of the dynamic environments. The agent is depicted as a red circle. The direction of rotation of the walls is shown with red arrows. The goal position is shown with ⋆.

Starting from a given state $s_0$ as the root node, the function `planRec` maintains and iteratively expands a tree structure, with states as nodes and actions as edges. During each iteration, the algorithm selects a path from the root to a leaf node. Upon reaching a leaf node, it extends the tree by sampling a number ($k$) of actions from the leaf, thereby adding new nodes and edges to the tree. The path selection process is based on the Upper Confidence Bound (UCB) formula [83], which utilizes an internal representation of the state-action values at each tree node (denoted by $\hat{Q} : \mathcal{S} \times \mathcal{A} \rightarrow \mathbb{R}$), to balance exploration of less frequently visited paths with exploitation of paths known to yield high returns. Once a path is selected, the algorithm updates the value of $\hat{Q}$ for each node on the selected path by backpropagating the accumulated rewards from subsequent nodes. These updates are averaged by each state-action's visit count to achieve more precise estimates.

After a specified number of iterations ($I$), the algorithm uses $\hat{Q}$ to greedily select an optimal path from the root to a leaf node and returns the corresponding sequence of actions as its final result.

## A.3 Additional Experimental Results

### A.3.1 Benchmarks

A detailed description of benchmarks used in our evaluation is presented below:

- `obstacle`: This benchmark involves a robot moving on a 2D plane, aiming to reach a goal position without colliding with a stationary obstacle. This obstacle is positioned to the side, impacting the robot only during its exploration phase and not on the direct, shortest route to the goal.

- `obstacle2`: This benchmark involves a robot moving on a 2D plane, aiming to reach a goal position without colliding with a stationary obstacle. This obstacle is positioned between the starting point and the goal, requiring the robot to learn how to maneuver around it.

- `mount-car`: This benchmark requires moving an underpowered vehicle up a hill without falling off the other side.

- `road`: This benchmark requires controlling an autonomous car to move in one dimension to a specified end position while obeying a given speed limit.

- `road2d`: This benchmark requires controlling an autonomous car to move in two dimensions to a specified end position while obeying a given speed limit.

- `dynamic-obs`: This benchmark features multiple non-stationary obstacles that move deterministically in small circles on the path from the agent to the goal.

- `single-gate`: In this benchmark, the goal position is surrounded by a circular wall with a small opening, allowing the agent to pass through. The position of the opening continuously rotates around the goal position. This is visualized in Fig. 6a.

- `double-gates`: This benchmark is similar to `single-gate`; however, the goal position is surrounded by two concentric rotating walls. This is visualized in Fig. 6b.

- `double-gates+`: This benchmark is similar to `double-gates`; however, the thickness of the rotating walls is increased. This poses a greater challenge, as the agent has a shorter

time window to cross through the opening without colliding with the rotating wall. This is visualized in Fig. 6c.

For the dynamic benchmarks above (`dynamic-obs`, `single-gate`, `double-gate`, and `double-gate+`), the action space of the double integrator agent consists of acceleration in the $x$ and $y$ directions. The action space for the differential drive agent consists of the torque value on the left wheel and the torque value on the right wheel.

Additionally, the observation space of benchmarks with a double integrator agent consists of the position and velocity vectors of the agent, while for a differential drive agent, the observation space includes position, velocity, and pose angle. The observation space for `single-gate`, `double-gate`, and `double-gate+` also contains an angle term for each wall, corresponding to the current rotation of each wall. The observation space in `dynamic-obs` additionally includes an angle term for each obstacle, indicating how far each obstacle has moved along its circular trajectory.

### A.3.2 Implementation

We implemented `DMPS` by modifying the Twin Delayed Deep Deterministic Policy Gradient (TD3) algorithm [15], following the description presented in Algorithm 1.

Each of the dynamic benchmarks were trained for 200,000 timesteps with a maximum episode length of 500. The static environments were trained for the number of timesteps prescribed by the sources of the environments. Namely, `mount-car` was trained for 200,000 timesteps with a maximum episode length of 999 [84], `obstacle` and `obstacle2` were trained for 400,000 timesteps with a maximum episode length of 200 [16], and `road` and `road2d` were trained for 100,000 timesteps with a maximum episode length of 100 [16]. Each experiment was run on five independent seeds. We used prior implementations of `REVEL` [16], `PPO-Lag` [85], and `CPO` [85] to run our experiments. Trained models were test evaluated every 10K timesteps, independently from the training loop. Each test evaluation consisted of 10 runs. The final test evaluation was used to determine the average per-episode return and per-episode shield invocation rate for that random seed. These are the values displayed in the tables from section 6.

The reward functions used in [16] for the static environments are non-standard, and based upon a cost-based framework of reward. In light of this, we used the canonical reward functions for our evaluations. For `mount-car`, we use the original reward function defined in [84]. For `obstacle`, `obstacle2`, `road`, and `road2d`, we use the canonical goal environment shaped reward function.

The negative penalty for safety violations in `TD3` was taken to be large enough so that the agent could not move through obstacles and still maintain positive reward. In most cases, the penalty was simply the negation of the positive reward incurred upon successfully completing the environment. The episode did not terminate upon the first unsafe action. For `CPO`, we reduced tolerance for safety violations by reducing the parameter for number of acceptable violations to 1.

Our experiments were conducted on a server with 64 available Intel Xeon Gold 5218 CPUs @2.30GHz, 264GB of available memory, and eight NVIDIA GeForce RTX 2080 Ti GPUs. Each benchmark ran on one CPU thread and on one GPU.

### A.4 Analysis of Computational Overhead

As mentioned in the main text, our implementation uses MCTS in an anytime planning setting, where we devote a fixed number of node expansions to searching for a plan, and we choose the best plan found at the end of the procedure. This bounds the clock time needed for planning linearly in the compute budget allocated. However, in the worst case, one may need a compute budget exponential in size with respect to the planning horizon $H$ if one wishes to explore the planning state densely. This is common to all general planners.

Here, we try to experimentally evaluate how the required compute budget scales as a function of the planning horizon. We re-evaluate the `double-gates+` environment (under double integrator dynamics). For each planning horizon $H$ in range $[2, 9]$, we count the average number of node expansions that MCTS needs before it successfully explores 10 states at depth $H$. We use this as a proxy for successful exploration of the horizon $H$ search space. The results of this experiment can be found in Figure 7. As expected, an exponential relationship emerges.

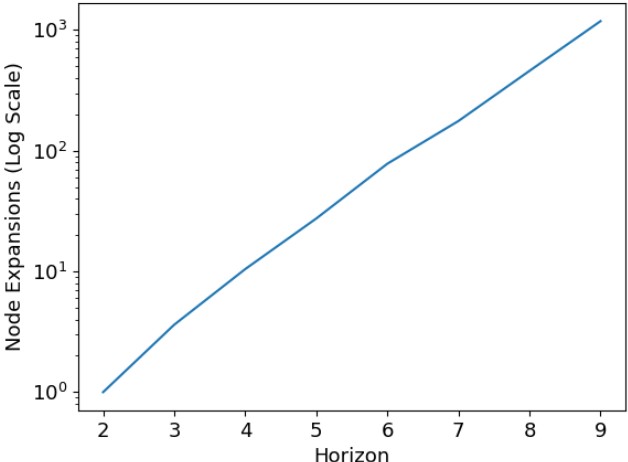

Figure 7: Planner Computation Scaling

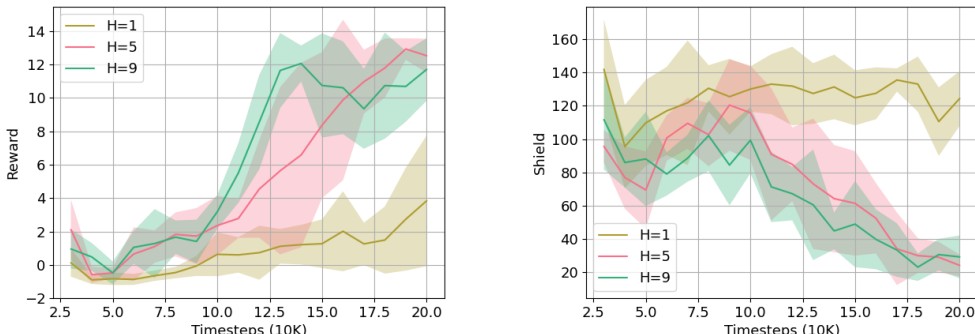

Figure 8: Multiple Horizons Experiments

## A.5 Effect of Planning Horizon

We expand on the question of whether small planning horizons are sufficient to solve tricky planning problems. Our algorithm is designed to ensure that the planner accounts for both short-term and long-term objectives. As detailed in section 5, the objective of the planner consists of two terms: 1) the discounted sum of the first $n$ rewards, and 2) the estimated long-term value of the penultimate step in the plan horizon, as determined by the agent's $Q$ function. The second part of the objective function is specifically included to ensure that the planner does not return myopic plans, and accounts for progress towards the long-horizon goal. Since the planner optimization objective includes this second term, even a small-horizon planner can output actions with proper awareness of long-horizon recovery events. The length of the horizon affects how close to the globally optimal solution the result is, with a tradeoff of computational cost, as was established in Theorem 5.1.

To see this empirically, we reevaluated the `double-gates+` environment (double integrator dynamics) with horizons of 1, 5, and 9. The graphs of attained reward and shield triggers from this experiment are shown in Figure 8.

Comparing the $H = 1$ and $H = 5$ agents, the $H = 5$ agent substantially outperforms the $H = 1$ agent in both shield triggers and reward. Comparing $H = 5$ and $H = 9$, the $H = 9$ agent reached high performance and low shield triggers faster than the $H = 5$ agent. However, both the $H = 5$ and $H = 9$ agents converge to the same performance eventually.

