# OpenReview forum: "Dynamic Model Predictive Shielding for Provably Safe Reinforcement Learning"
_NeurIPS.cc/2024/Conference — NeurIPS 2024 poster_

### Official Review · Reviewer_d437 · 2024-07-04

**Soundness:** 2
**Presentation:** 3
**Contribution:** 3
**Rating:** 5
**Confidence:** 3

**Summary:**

Naive model predictive shielding may overly restrict exploration thereby preventing an RL agent from learning a policy with good performance. In order to prevent this, the authors propose a method to optimise a backup policy that is provably safe using an online planner. An approximate model such as double integrator or differential drive is used for planning. Improvements are demonstrated on five benchmarks that involve static or dynamic obstacle avoidance as compared to provably safe and approximately safe RL methods. A provable guarantee is provided for recovery regret.

**Strengths:**

Presentation is clear with backing proofs and demonstrable results

Problem that is being solved is clearly delineated and addressed using sound techniques

Experimental comparisons are performed rigorously with attention to detail

**Weaknesses:**

Literature review and comparisons are partial to the RL literature. There is a long-standing literature in control [A, B, C] to use an approximate model to plan using predictive control. A whole host of methods to learn a safety-filter/shielding on the fly has been explored with robust optimization-based offline and online control techniques. Most of these methods would implicitly solve the problem this paper is trying to address. However, it is interesting that the paper uses the Q function in the online optimization. This aspect is novel and unique to this paper.

It is unclear how much computation and time it takes to run MCTS online at each time in order to do dynamic shielding at runtime.

Dynamics model such as double integrator and differential drive are too simple. It would be interesting to see how well these would work with more complicated and/or higher-dimensional dynamics.

[A] Breeden, Joseph, and Dimitra Panagou. "Predictive control barrier functions for online safety critical control." 2022 IEEE 61st Conference on Decision and Control (CDC). IEEE, 2022.

[B] Wabersich, Kim P., and Melanie N. Zeilinger. "Predictive control barrier functions: Enhanced safety mechanisms for learning-based control." IEEE Transactions on Automatic Control 68.5 (2022): 2638-2651.

[C] Wabersich, Kim P., et al. "Data-driven safety filters: Hamilton-jacobi reachability, control barrier functions, and predictive methods for uncertain systems." IEEE Control Systems Magazine 43.5 (2023): 137-177.

**Questions:**

Have the authors explored the space of RL training algorithms and methods to test this approach?

Are there any advantages of using DMPS if the performance policy is not using RL and uses imitation learning or no learning at all? Exploration is important only for RL.

**Limitations:**

The authors have discussed limitations and there is potential for the approach to scale even though the experiments in the paper are on simple examples.

---

> ### Author Rebuttal · Authors · 2024-08-06
>
> We thank the reviewer for their many insightful comments and suggestions. We respond to their questions and concerns below.
>
> **Adding traditional control and non-RL methods to literature review**
>
> We appreciate the reviewer's paper recommendations and will include them, along with classical control papers, in our literature review.
>
> We did not compare our approach to non-RL methods experimentally. One of RL's strengths is its generality—agents can satisfy complex specifications with just an abstract environment and a simple reward signal. While this generality is valuable, specialized methods often outperform RL in specific tasks. Since our paper is primarily RL-focused, we believe that the fairest comparison is against other RL approaches.
>
> **Concerns over simplicity of the environment dynamic models**
>
> Since the planner and RL algorithm treat environment dynamics as black boxes, more complex dynamics don't necessarily challenge the agent, as long as they are deterministic and allow sufficient range of motion for task completion. We agree that increasing the dimensionality of the dynamics would make the task harder for both the RL algorithm and the planner and believe that this would be an interesting avenue for future research.
>
> **Using other RL training algorithms**
>
> Our algorithm can generally work with any off-policy RL algorithm. We briefly explored using DDPG and SAC as the training algorithm and did not find reliable improvement.
>
> **Advantages of using DMPS if performance policy does not use RL (instead using IL or no learning)**
>
> RL-based approaches like DMPS can learn from interactions with the environment, allowing them to explore states and actions essential for model learning. Unlike imitation learning, which often suffers from distribution shift, RL methods are more resilient. Moreover, RL methods are more general than alternatives and need minimal input. Imitation learning requires expert trajectories and sometimes an expert controller, which can be costly. Classical learning-free methods rely on environment-specific assumptions. We believe that the generality of RL makes it a valuable paradigm on its own.

---

> ### Comment · Reviewer_d437 · 2024-08-12
> **Thank you for the rebuttal**
>
> I read the author response and other reviews. Overall, the paper is good but I would still consider it a borderline accept.
>
> For me, the computational cost of running the local planner in real-time and the divergence of the planned trajectories from reality due to imperfect models are still a concern. MCTS is usually distilled into a neural network so that the safety recovery policy can be run in real-time. I am not sure whether that is appropriate here.
>
> Further, I did not go through the proof in the Appendix in detail.

---

> > ### Author Response · Authors · 2024-08-13
> >
> > We thank the reviewer for their response.
> >
> > We agree with the reviewer that the computational cost of running the local planner poses a potential concern for running the model in real-time. There have been prior works applying highly optimized planning to much larger horizons while running real-time on physical robots. For instance, [5] uses a MPPI planner that explores upto a horizon of 100 while running on a real robot.
> >
> > Insofar as the plan horizon in our design makes real-time operation prohibitively expensive, we note that at test time, one can truncate the planner horizon to be much lower than at train time, thus allowing the model to operate efficiently in real time. At the end of training, the agent has an accurate Q function, which allows even a short-horizon planner to properly optimize the agent's infinite-horizon return.
> >
> > To see this empirically, we took the final DMPS models (trained with an H=5 planner) from the double-gates+ environment and evaluated them with truncated horizon. Using just an H=1 planner, our model received a reward of 10.86. Recall from the graph in our global rebuttal that an agent **trained** on double-gates+ with an H=1 planner only received reward of 2.9. The jump in performance demonstrates both the necessity of a non-trivial horizon at train time and the sufficiency of a trivial horizon at test time. Using an H=2 planner at test time gives a reward of 12.47, which closely approximates the H=5 planner (as used in training) reward of 13.0.
> >
> > The reviewer's concern about the divergence of planned trajectories from reality is also valid. Our work uses analytic models, and these have the potential to be inaccurate. Learning more accurate models is an orthogonal research direction that has been studied prior [1,2,3,4]. An interesting direction for follow-up work could investigate combining this line of research with our own.
> >
> > [1] Atreya, P., Karnan, H., Sikand, K. S., Xiao, X., Rabiee, S., & Biswas, J. (2022). High-Speed Accurate Robot Control using Learned Forward Kinodynamics and Non-linear Least Squares Optimization. IROS 2022.
> >
> > [2] Chua, K., Calandra, R., McAllister, R., & Levine, S. (2018). Deep Reinforcement Learning in a Handful of Trials using Probabilistic Dynamics Models. NeurIPS 2018.
> >
> > [3] Karnan, H., Sikand, K. S., Atreya, P., Rabiee, S., Xiao, X., Warnell, G., … Biswas, J. (2022). VI-IKD: High-Speed Accurate Off-Road Navigation using Learned Visual-Inertial Inverse Kinodynamics. IROS 2022.
> >
> > [4] S. Piche, B. Sayyar-Rodsari, D. Johnson, and M. Gerules, “Nonlinear model predictive control using neural networks,” IEEE Control
> > Systems Magazine, vol. 20, no. 3, pp. 53–62, 2000.
> >
> > [5] Williams, G., Wagener, N., Goldfain, B., Drews, P., Rehg, J. M., Boots, B., & Theodorou, E. A. (2017). Information theoretic MPC for model-based reinforcement learning. ICRA 2017.

---

### Official Review · Reviewer_jAJK · 2024-07-09

**Soundness:** 3
**Presentation:** 4
**Contribution:** 4
**Rating:** 7
**Confidence:** 3

**Summary:**

This paper proposes a new method for safety shielding. More precisely, the authors extend Model Predictive Shielding (MPS), where an agent reverts to a safe backup policy if, for the next predicted state, this policy would not be able to guarantee safety anymore. MPS is often overly conservative, particularly in cases where the backup policy is very different from the final policy (for example, it may only consider breaking, while the final policy may be able to steer around an object). To improve this, whenever an action is potentially unsafe, the agent first uses a short-horizon planner to see if there exists some safe action that may be better than the one of the backup policy (i.e., one for which the backup policy could still recover in the future, but for which our learned agent predicts a higher reward). The authors formalize this framework and show recovery regret for this framework diminishes exponentially with the horizon. Next, they show that an implementation of this framework outperforms prior methods, both in terms of performance and the number of required shield invocations.

**Strengths:**

The topic of the paper, safety shielding, is relevant and significant. Safe RL (and particularly, safety shielding) is a promising line of research but is often overly conservative in practice: the methods proposed in this paper take a step toward reducing this problem while still giving formal guarantees about safety. The topic is relevant for the NeurIPS community (particularly those interested in RL), both as a method that could immediately be used or to extend the method to more complex settings (i.e., with a stochastic/unknown model).

The paper is well-written and easy to read: the intuition behind the method is clear, and the analysis of the results is easy to follow. The framework is well formalized (using visualizations where helpful), and the given pseudo-code helps with reproducibility. The experiments are extensive and convincingly show the advantages of the proposed method.

**Weaknesses:**

Apart from some minor remarks that I add below, this paper has one main weakness: it does not clearly indicate the computational complexity of its method nor the scalability. The results do not show computation times, and (as far as I could tell) no mention is made of either the average planning time or some time limit for this planning phase. From some ball-parking, the additional time required for this method may be significant (solving up to millions of short-horizon planning problems), so a quantification of this computational cost should be provided.

Some more minor remarks:
* The paper only mentions how the framework is implemented (i.e., what RL & planning method it uses) in the appendix: it would be nice to (briefly) mention this in the results section as well;
* In Table 2, the results of CPO and TD3 are not bold, even though some are equal to those of the best frameworks: this should be fixed;
* One limitation of the proposed framework is that it assumes the environment is deterministic: it would be nice to mention this in the limitations section.

**Questions:**

As mentioned in the 'weaknesses' section, I have one main question: how does the computational complexity of your method compare to those of the benchmarks, particularly MPS? I will change my rating if this question is not adequately answered.

**Limitations:**

Limitations are adequately addressed.

---

> ### Author Rebuttal · Authors · 2024-08-06
>
> We thank the reviewer for their many insightful comments and suggestions. We respond to their questions and concerns below.
>
> **Questions over computational complexity**
>
> We analyze the question of the planner’s computational cost in more depth in the global rebuttal. We restate a short summary of the global response here.
>
> There is a tradeoff between the quality of the recovery plan, and the computational cost incurred in the planner searching for it. The look-ahead controls this tradeoff. In our implementation, we use MCTS in an anytime planning setting, where we devote a fixed number of node expansions to searching for a plan, and we choose the best plan found at the end of the procedure. The clock time needed would simply be linear in the allocated compute budget. However, the worst-case computational complexity to densely explore the search space, as in general planning problems, would be O(exp(H)) where H is the look-ahead length, since the planning search space grows exponentially. Since the remaining baselines do not do any kind of planning, they use constant time per timestep, though this comes at an optimality cost.

---

> > ### Comment · Reviewer_jAJK · 2024-08-09
> >
> > Thanks to the authors for their extensive answers. Based on the other reviews and rebuttals, I have two comments:
> >
> > Firstly, thanks for the detailed explanation on the computational complexity. I think the details about your particular implementation of the planner (i.e., using a Python implementation of MCTS with a node budget of 100 and horizon 5, leading to planning times of ~0.4s per timestep) are relevant to include in the experimental section of the paper to give readers an idea of the computation cost of using your method.
> >
> > Secondly, I found your analysis of the effect of the planning horizon on performance interesting to read. Although you explain that the effect of the horizon on performance is limited, I still agree with reviewer BFvm that choosing a horizon that balances computational costs with performance could be tricky in more complex environments. I think this is worth adding to the limitation section of the paper.
> >
> > I'd like to hear from the authors if they plan to make these changes in the revised version of the paper. Otherwise, I have no further questions or comments.

---

> > > ### Author Response · Authors · 2024-08-10
> > >
> > > We thank the reviewer for highlighting the point regarding the effect of the planning horizon on performance. We will expand on this in the limitations section and incorporate the key discussion points (choosing a horizon that balances computational costs with performance could be tricky in more complex environments), and will add the detailed analyses and graphs to the appendix to provide further information on this topic. We will also add the highlighted implementation details in the Experiments section.

---

### Official Review · Reviewer_NRWz · 2024-07-10

**Soundness:** 3
**Presentation:** 4
**Contribution:** 3
**Rating:** 7
**Confidence:** 5

**Summary:**

The authors introduce Dynamic Model Predictive Shielding (DMPS) an extension of Model Predictive Sheilding (MPS) that adress some of its key limitations, such as overconservatism when deploying the backup policy which consequently hinders exploration of the neural 'task' agent and slows down conergence. The key innovation of DMPS is that it incoropoates a local planner for dynamic recovery that leverages both a pre-computed backup policy and the neural 'task' policies Q values for planning for short and long horizon returns while maintaining safety of the system. DMPS is a provably safe reinforcement learning (PSRL) method, meaning that it guarantees safety of the system (regardless of the underlying neural 'task' policy) by only exploring within the set of safe and recoverable states defined by the backup policy.  This realised by planning for a fixed n step trajectory and checking whether the system can be recovered by the backup policy given the action proposed by the agent. The authors demonstrate that DMPS outperforms MPS and other baselines in terms of performance and safety in various benchmarks. It also emphasizes the importance of aligning reinforcement learning agents with real-world safety requirements, while discussing some of the limitations of their approach.

**Strengths:**

The paper has several strengths: I find that the paper is very well written and easy to follow, with sufficient details in necessary places and abstractions in other places where the details may not immediately matter, as such, it is a very nice read. The theoretical analysis of the recovery regret is convincing and interesting. Furthermore, the overall framework is attractive from the point of view that it is provably safe, something I personally find is crucial for deploying RL in the real world, rather than safe at convergence or in expectation like a lot safe RL methods. I find that the dynamic planning module is an innovative solution to the intuitive issue faced by most shielding methods (Figure 2) and I feel that this work constitutes a step in the right direction for improving shielding methods and making them more practical. The experimental evaluation I feel is strong and thorough as in most cases DMPS clearly outperforms MPS and REVEL, although I think it is missing something (see weaknesses).

**Weaknesses:**

The key weakness of the PSRL framework is the reliance on a perfect (or sufficiently accurate) dynamics model of the environment, the safety performance of the backup policy and the computation of the safe invariant set. In contrast to the first shielding approaches for RL [1], which operate primarly on discrete state and actions spaces, DMPS does not need to compute the shield before learning can begining which significantly reduces the engineering overhead before training. This of course comes at a cost, in practice the shields in [1] are substatially more lightweight during "inference", (although in theory there could be exponential blow up) in part due to only operating on discrete or discretized state/action spaces but also because a lot of the necessary computation is done before hand. This is a key limitation of DMPS as it relys on planning at each timestep which might be costly and infeasible for on-board computation or edge devices. Fruthermore, it seems that there is still a significant amount of prior knowledge required for DMPS to work effectively, first we have to have a "perfect" dynamics model (for provable guarantees) secondly I presume we need to handcraft a backup policy and then compute its invariant safe set so as to plan for recoverability. The first limitation is mentioned in the paper but not really discussed in much detail, the second limitation is find is crucial and I don't think is really mentioned in the paper. In particular it is a non-trivial challenge to come up with a backup policy that has a maximal safe invariant set, perhaps for the environments the authors consider it is easy (just decelerate) but for more dynamics environments and in general this is not the case and I feel like more discussion about both these limitations (i.e. the limitations of the PSRL setting) is needed.

While I find the experimental evaluation compelling I feel it is slightly misleading and it is missing something. In Table 2 CPO and TD3 score the same or higher in a few of the static benchmarks but there scores are not in bold, is there a reason for this that I am missing? I also feel like a comparison to PPO-Lag or DDPG-Lag would really help make the results that bit more convincing.

All that being said, in principle I advocate for acceptance of this paper.

[1] Alshiekh, Mohammed, et al. "Safe reinforcement learning via shielding." Proceedings of the AAAI conference on artificial intelligence. Vol. 32. No. 1. 2018.

**Questions:**

Most of my questions are technical:

For each of the enviroments you consider how are the backup policies constructed and how are the invariat safe sets determined?

For each of the environments what are the maxmimum number of steps needed to deccelerate the vehicle to zero or avoid obstacles and is your choice of n=5 sufficient?

What would be suitable ways of modelling the environment from experience to obtain uncertainty estimates, for example would Gaussian Process modelling suffice?

Do you assume any observation noise or just perfect access to the current state, if not how would you incorporate this into your framework?

**Limitations:**

See weaknesses.

---

> ### Author Rebuttal · Authors · 2024-08-06
>
> We thank the reviewer for their many insightful comments and suggestions. We respond to their questions and concerns below.
>
> **Construction of backup policies and determination of the invariant sets**
>
> In static environments, the backup policy involves braking as hard as possible. In dynamic environments, the agent checks if it is in an obstacle's trajectory; if so, it accelerates away, otherwise, it brakes. The backup policy is straightforward, aiming to halt the agent quickly. It can be determined through basic understanding of environment dynamics, or by training a neural backup policy.
>
> The Model Predictive Shielding (MPS) framework automatically synthesizes the invariant safe set from the backup policy. The MPS framework defines “stable states” where safety is guaranteed. In static environments, stable states are those where the agent has zero velocity; in dynamic environments, the agent must have zero velocity and be outside the trajectory of all obstacles. A state s is within the invariant safe set ($S_{rec}$) if a trajectory from s to a stable state can be found by forward-simulating the backup policy without violating safety conditions.
>
> The invariant set's construction is automatic, requiring no manual engineering, and is generally sufficient for most safety problems. States not within $S_{rec}$​ that could allow safe navigation are usually undesirable, as they are too close to obstacles for braking to be safe.
>
> **Maximum number of steps needed to decelerate the vehicle and sufficiency of n=5**
>
> The planning horizon doesn't need to cover all steps to decelerate the vehicle to zero. Each state in MCTS expansions is verified to be in $S_{rec}$ through forward simulation, ensuring every state in the search is safe. This simulation isn't part of the planning horizon.
>
> For context, it takes a maximum of 10 timesteps to decelerate to a stop in both the differential drive and double-integrator dynamics.
>
> **Suitable ways to model environment noise**
>
> We assume the environment is deterministic with no observation noise. However, in RL, Gaussian modeling can effectively handle observational uncertainty, as shown in [2]. More complex models, like uncertainty-aware deep learning, have also been explored [1].
>
> **Experiments**
>
> We thank the reviewer for their comments on the experimental results. We have included PPO-lag as a baseline and corrected the bolded numbers in the tables. The tables are attached in the global rebuttal.
>
> [1] Chua, K., Calandra, R., McAllister, R., & Levine, S. (2018). Deep Reinforcement Learning in a Handful of Trials using Probabilistic Dynamics Models. In S. Bengio, H. M. Wallach, H. Larochelle, K. Grauman, N. Cesa-Bianchi, & R. Garnett (Eds.), Advances in Neural Information Processing Systems 31: Annual Conference on Neural Information Processing Systems 2018, NeurIPS 2018, December 3-8, 2018, Montréal, Canada (pp. 4759–4770). Retrieved from https://proceedings.neurips.cc/paper/2018/hash/3de568f8597b94bda53149c7d7f5958c-Abstract.html
>
> [2] Kaufmann, E., Bauersfeld, L., Loquercio, A., Müller, M., Koltun, V., & Scaramuzza, D. (2023). Champion-level drone racing using deep reinforcement learning. Nat., 620(7976), 982–987. doi:10.1038/S41586-023-06419-4

---

> > ### Comment · Reviewer_NRWz · 2024-08-13
> >
> > Thanks for these insights they are really helpful, and thanks for including PPO-Lag in your experiments. I see this now as quite a strong paper and will raise my score to 7.
> >
> > Final question: the safe invariant set $S_{rec}$ is this computed automatically before training (with the backup policy) or is this computed on-the-fly by the dynamic planner?

---

> > > ### Author Response · Authors · 2024-08-13
> > >
> > > We thank the reviewer for their response and for raising their score! The invariant set is computed on-the-fly, though the overhead for checking admittance in $S_{rec}$ is lightweight for individual states.

---

### Official Review · Reviewer_xs2Y · 2024-07-11

**Soundness:** 3
**Presentation:** 3
**Contribution:** 3
**Rating:** 6
**Confidence:** 4

**Summary:**

The approach called dynamic model-predictive shielding for safe reinforcement learning is proposed as an improvement over its static counterpart. The main idea is to optimize for expected return on action with respect to the reinforcement-learning task when choosing a shielding backup action, and to incorporate planning horizon prediction into learning for the policy to learn to avoid unsafe actions. This dynamic version is evaluated on several static and dynamic obstacle-avoidance benchmarks and compared to static model-predictive shielding and three more planning-based approaches.

**Strengths:**

The core idea of the approach is interesting and potentially valuable: to achieve synergy between safety and optimal performance in model-predictive shielding via incorporating planning into policy learning and taking expected performance into account during backup planning. Similar attempts have been done previously. In comparison, this work proposes a novel notion of "recovery regret" as a heuristic to guide mutual integration of planning and reinforcement learning.

The strength of the paper is in extensive evaluation and comparison to multiple approaches. The notion of recovery regret can also be of independent interest for model-predictive shielding research. Dynamic shielding outperforms other approaches in the evaluation in terms of the number of shielding invocations, which indicates synergy between planning and learning over time.

**Weaknesses:**

Potential weaknesses of the approach are in scalability of the planner and tightness of the probabilistic bounds on safety.

Minor:
- "more optimal recovery approach" --> an optimal/a better

**Questions:**

Questions
1. In Figure 1, what are green and red lines, and a blue blob?
2. How does the local planner scale with respect to the look-ahead?
3. Does the local planner have to recompute the look-ahead prediction every time it is invoked or does it reuse previous results if the agent continues along the same trajectory?
4. What is the overhead of the planner's computations?
5. How does the planning limit affect safety and optimality guarantees?
6. MCTS typically struggles to plan for overly constrained problems and complex planning tasks. How does the approach scale with respect to the planning task complexity?

**Limitations:**

The authors explicitly discuss limitations which are fairly common for problem domain.

---

> ### Author Rebuttal · Authors · 2024-08-06
>
> We thank the reviewer for their many insightful comments and suggestions. We respond to their questions and concerns below.
>
> **Figure 2 clarification**
>
> We assume that the reviewer’s question is referring to Figure 2. Figure 2 demonstrates the example described in the text, and in particular, part (d) of the figure is discussed in Section 5 (Lines 208-215). The red and green lines represent two locally optimal paths in the planning tree. The blue blob represents a water puddle that would result in lower returns if crossed. The planner selects the green planner here, informed by its access to the long-term $Q$ function.
>
> **Reusing lookahead predictions in the local planner**
>
> The planner is only used when the neural policy suggests an action that cannot be certified as recoverable. Reusing prior results is beneficial only if the planner is invoked consecutively due to the base policy acting recklessly. In such cases, previous search results can be reused, similar to sub-tree reuse in the POMCP algorithm [2]. Our current implementation does not include this feature, but it can be easily added.
>
> **Planner overhead**
>
> Beyond the computational cost of running the planner (discussed above), there is minimal overhead in our implementation. To minimize the overhead, we designed the planner to leverage the environment's simulated transition function so that the state and action space between the planning problem and the RL formulation is identically the same. Thus, when DMPS needs to invoke the planner, it just initializes a planning environment with the current world state, and the planner directly queries the transition function internally during its search.
>
> **Effect of planning limit on safety and optimality guarantees**
>
> When using a planner with proven completeness (or probabilistic completeness) properties, DMPS can always guarantee a safe recovery plan, given enough planning time. However, due to finite computational limits, the planner might not find a solution in a fixed time. Thus, DMPS can rely on its fallback policy when the planner fails to provide a valid solution (branch 4 in Figure 1). As such, the incorporation of a planner in DMPS does not affect the provable safety guarantees. In practice, we find that DMPS never uses the fallback policy in our experimental settings.
>
> The relationship between the optimality guarantee and the planning depth is outlined in Theorem 5.1, with more specific statements and proofs given in Appendix A.1. The planner’s regret decays exponentially as a function of the planning depth.
>
> **Scaling of approach with respect to planning task complexity**
>
> As the reviewer noted, MCTS struggles with overly constrained problems, making the choice of planner crucial. We used MCTS in our evaluation due to its wide applicability, but some domains may have more effective planners. For instance, Informed RRT* excels at motion planning in constrained environments with dense clusters [1].
>
> [1] Gammell, J. D., Srinivasa, S. S., & Barfoot, T. D. (2014). Informed RRT*: Optimal sampling-based path planning focused via direct sampling of an admissible ellipsoidal heuristic. 2014 IEEE/RSJ International Conference on Intelligent Robots and Systems, IROS 2014, Chicago, IL, USA, September 14-18, 2014, 2997–3004. doi:10.1109/IROS.2014.6942976
>
> [2] Silver, D., & Veness, J. (2010). Monte-Carlo Planning in Large POMDPs. In J. D. Lafferty, C. K. I. Williams, J. Shawe-Taylor, R. S. Zemel, & A. Culotta (Eds.), Advances in Neural Information Processing Systems 23: 24th Annual Conference on Neural Information Processing Systems 2010. Proceedings of a meeting held 6-9 December 2010, Vancouver, British Columbia, Canada (pp. 2164–2172). Retrieved from https://proceedings.neurips.cc/paper/2010/hash/edfbe1afcf9246bb0d40eb4d8027d90f-Abstract.html

---

> > ### Comment · Reviewer_xs2Y · 2024-08-09
> >
> > Thank you for the detailed answers and proposed extensions. It would be beneficial to include this discussion into the paper. I have no further questions.

---

> > > ### Author Response · Authors · 2024-08-10
> > >
> > > We thank the reviewer for their acknowledgment. We plan to expand the limitations section with points from the Rebuttal and add the detailed analyses and graphs to the Appendix to provide further information.

---

### Official Review · Reviewer_BFvm · 2024-07-12

**Soundness:** 3
**Presentation:** 3
**Contribution:** 2
**Rating:** 5
**Confidence:** 4

**Summary:**

The paper seeks to address provably safe RL problems where safety must be ensured even during training. It proposed DMPS, which enhances prior Model Predictive Shielding approach, to dynamically select safe actions when danger is imminent. DMPS employs local planner to plan for recovery actions and the planner objective consists of both short-term and long-term rewards. Feedback from the planner can then be used to incrementally train the neural policy to guide it towards safe policy set.

**Strengths:**

1. Quality
* Overall, the approach described in the paper is sound and it combines many established components (e.g. backup policy, local planner, estimate future reward using model unrolling and Q-estimate) to facilitate safe RL.
* The paper provides theoretical bound on the recovery regret as the sampling limit in the local planner approaches inifinity.
2. Clarity
* The paper is written in a clear and lucid manner. The figures, algorithm and equations are structured in a way that is easily understandable to the readers.

**Weaknesses:**

1. Originality
*  The main difference between DMPS and MPS is the use of local planner when backup policy is triggered. The technical approach used in DMPS is not particularly new as there are already some similar approaches of estimating a safety Q-value and perform planning based on the Q-value [1, 2].
2. Significance
* The only difference between DMPS and the prior MPS seems to be the local planner and (as discussed in point 1) this local planner is not particularly novel. Having said that, I do agree that the proposed DMPS does show improvement over MPS in some experiment scenarios.
* While the paper mentions a small planning horizon is sufficient for the local planner to plan safe yet rewarding actions, I feel that this may not be true in most cases. To steer the agent back to safety (and yet rewarding), a long sequence of actions may be required. If the planning horizon is set too small, then DMPS falls back to backup policy and the performance would be the same as MPS. In this case, I guess the only solution is to increase in planning horizon and in turn increase the computational overhead of DMPS exponentially.
* The local planner requires perfect information of the transition and the transition must be deterministic. This may restrict its applicability, especially given that there're prior work on model-based RL where transition can be stochastic and learned instead.

References
[1] Clavera, I., Fu, Y. and Abbeel, P., Model-Augmented Actor-Critic: Backpropagating through Paths. In International Conference on Learning Representations.
[2] Thomas, G., Luo, Y. and Ma, T., 2021. Safe reinforcement learning by imagining the near future. Advances in Neural Information Processing Systems, 34, pp.13859-13869.
[3] Nagabandi, A., Kahn, G., Fearing, R.S. and Levine, S., 2018, May. Neural network dynamics for model-based deep reinforcement learning with model-free fine-tuning. In 2018 IEEE international conference on robotics and automation (ICRA) (pp. 7559-7566). IEEE.
[4] Chua, K., Calandra, R., McAllister, R., and Levine, S. Deep reinforcement learning in a handful of trials using probabilistic dynamics models. In Advances in Neural Information Processing Systems. 2018.
[5] Janner, M., Fu, J., Zhang, M. and Levine, S., 2019. When to trust your model: Model-based policy optimization. Advances in neural information processing systems, 32.

**Questions:**

1. In the first sentence of Section 6.2, you mentioned that the total return is averaged over the last 10 training episodes. Are you evaluating them using the same (single) random seed and only use the last 10 training episodes for evaluation?

2. Given that both the states and actions are continuous, how do you apply MCTS to the local planner?

3. TD3 only maximizes a single reward objective. In your experiments, I guess you performed some sort of reward shaping for it to balance between safety and reward. Can you elaborate how do you incorporate safety into its objective and is there any weighting used?

4. Similarly for CPO, how do you incorporate safety into it? Do you specify a safety violation constraint?

5. (Related to Qn 3 & 4) I am surprised that TD3 and CPO rapidly overfits to conservative policy in dynamic environment. What do you think is the reason and is the weighting between safety and reward dynamically tuned?

**Limitations:**

The paper discussed the known environment model as its limitation. I agree that this is a limitation which warrants further investigation. As studied in [5], it is very challenging for a model to accurately predict future trajectories with long horizon. Since DMPS relies on having an accurate environment model for safety adherence, this may limit its applicability to practical scenarios where environment model is not given and needs to be learned.

Another related point is that it is unclear what value to set for the local planner horizon. Different tasks may require corrective action sequence of different lengths. Setting the horizon too short may revert DMPS performance back to MPS and setting the horizon too long may increase the computational overhead beyond acceptable threshold.

---

> ### Author Rebuttal · Authors · 2024-08-06
>
> We thank the reviewer for their many insightful comments and suggestions. We respond to their questions and concerns below.
>
> **Concerns over originality and significance of DMPS**
>
> The main novelty of our approach is the synergistic relationship between a local planner tasked with finding good recovery paths around unsafe regions and a neural policy that can learn from the planner's recovery strategies to navigate unsafe regions on its own. We thank the reviewer for highlighting the related papers (which we will cite in the revision), but note that both papers take quite different approaches from DMPS:
>
> The first paper does not deal with safe navigation, focusing instead on performance optimization in general RL settings. In that work, the training of the policy is disconnected from the planner; they first train a policy using their MAAC algorithm and directly insert it into an MPC planner. However, in the PSRL setting, adding a planner after the fact limits the performance of the policy since it was not trained to avoid safety-violating scenarios while simultaneously optimizing the task objective. The synergistic relationship between planner and policy in our approach avoids this problem.
>
> The second paper deals with safe navigation, but it does not use planning, instead taking a model-based RL approach. The key insights of this paper are dynamically adjusting the negative penalty for safety violations such that the penalty can "carry over" a discounted horizon, and using a model to synthetically generate more rollouts in training.
>
> **Reporting of average return in the experimental section**
>
> The mean is computed with randomized seeds. We take five random seeds, and for each random seed, we compute the mean return over the last ten training episodes. We then report the average score over the five random seeds. The standard deviation reported is across the seeds, not the episodes. We will clarify this in the text.
>
> **Using MCTS in continuous settings**
>
> We apply MCTS to the continuous domain following a strategy similar to previous works such as [2,3]. We deal with a continuous action space having a branch factor hyperparameter $B$, and then sampling $B$ continuous actions from the action space for each node expansion.
>
> **Reward shaping in TD3 to incorporate safety priorities**
>
> We tested two variants: one where TD3 received a negative reward penalty and ended the episode upon the first unsafe action, and another where each unsafe action incurred a penalty. The former led to frequent crashes, so we chose the latter. We set the penalty magnitude equal to the positive reward for completing the environment. Hyperparameter tuning showed that reducing the penalty too much led the policy to ignore safety constraints, while beyond a certain point, the penalty magnitude had little effect on behavior.
>
> **CPO safety parameters**
>
> CPO has a hyperparameter specifying the maximum tolerable number of safety collisions. Given our goal is provably safe RL, we set this parameter to 1 following the methodology of REVEL, [1]. We also tested higher tolerance values to ensure CPO wasn't hindered by the safety constraint but saw no performance improvement, so we maintained our original setting.
>
> **Overfitting of TD3 and CPO**
>
> In dynamic environments, reaching the goal requires safely avoiding obstacles. Hyperparameter tuning showed that low penalties for unsafe actions lead agents to tolerate collisions, while high penalties make them overly conservative. This indicates that avoiding obstacles while simultaneously achieving task goals is complex and can't be learned through reward signals alone.
>
> We did not use dynamic tuning. In CPO, the safety constraint is separate from the reward, so dynamic tuning was not possible. In TD3, changing the reward function mid-training is theoretically unsound and would likely destabilize the learned Q function, as it minimizes loss using a replay buffer with rewards spanning 10^6 timesteps.
>
> [1] Anderson, G., Verma, A., Dillig, I., & Chaudhuri, S. (2020). Neurosymbolic Reinforcement Learning with Formally Verified Exploration. NeurIPS 2020
>
> [2] Hubert, T., Schrittwieser, J., Antonoglou, I., Barekatain, M., Schmitt, S., & Silver, D. (2021). Learning and Planning in Complex Action Spaces. ICML 2021
>
> [3] Rajamäki, J., & Hämäläinen, P. (2019). Continuous Control Monte Carlo Tree Search Informed by Multiple Experts. IEEE Trans. Vis. Comput. Graph., 25(8), 2540–2553.

---

> ### Comment · Reviewer_BFvm · 2024-08-13
>
> I'd like to thank the authors for providing a detailed and to-the-point rebuttal.
>
> The two papers I quoted [1, 2] are part of safe RL literature and not in PSRL domains. The domains addressed aren't necessarily the same as DMPS but I think it does show that the usage of safety Q-value and look-ahead have been well investigated and aren't particularly original.
>
> After reading the global response on sufficiency of H=5, I still think that small planning horizon may not be sufficient for most cases. While it may be true in the domains tested in this paper where a shorter sequence of actions is sufficient to ensure safety, other types of domains may require longer sequence of actions to steer agent towards safety (yet rewarding) region. Increasing the look-ahead horizon in this case may increase the computational overhead exponentially.
>
> However, the authors did provide sufficient references showing that their setting (perfect information of transition dynamics and the transition must be deterministic) is common in PSRL literature. Since this work is likely to benefit the PSRL community, I'm willing to increase my final rating.
>
> UPDATE: I've increased the final rating in the Official Review.

---

> > ### Author Response · Authors · 2024-08-13
> >
> > We thank the reviewer for their valuable feedback and we will revise the paper accordingly. We also thank the reviewer for raising their score!

---

### Author Rebuttal · Authors · 2024-08-06

We thank the reviewers for their insightful feedback. We summarize the responses to common questions.

**Computational cost of the planner**

There is a tradeoff between the quality of the recovery plan, and the computational cost incurred in the planner searching for it. The look-ahead controls this tradeoff. In general, DMPS can leverage any planner available for the domain, and the computational cost of the planner with respect to the look-ahead will depend on the exact choice of the planner, and its hyperparameters.

In our implementation, we use MCTS in an anytime planning setting, where we devote a fixed number of node expansions to searching for a plan, and we choose the best plan found at the end of the procedure. The clock time needed would simply be linear in the allocated compute budget. However, the worst-case computational complexity to densely explore the search space, as in general planning problems, would be O(exp(H)) where H is the look-ahead length, since the planning search space grows exponentially.

Our implementation used MCTS with 100 node expansions, a plan horizon of 5, and a branch factor of 10, which amounts to exploring 1000 states in total. On average, we found that when the shield is triggered, planning takes 0.4 seconds per timestep. The code we used is unoptimized, written in Python, and single-threaded. Since MCTS is a CPU-intensive search process, switching to a language C++ would likely yield significant speed improvements, and distributing computation over multiple cores would further slash the compute time by the number of assigned cores.

Figure 1 shows the number of simulator state expansions needed by a search-based planner as a function of horizon length in the double-gates+ (double integrator) environment. Each node expansion requires equal computation time, hence this graph illustrates how the computation cost of the planner scales as a function of horizon length. Since we use an anytime planner, the number of actual states expanded (and hence the computational cost) is capped by a hyperparameter that controls the planner computation budget.

**Sufficiency of planning horizon H=5**

Our algorithm is designed to ensure that the planner accounts for both short-term and long-term objectives. As detailed in Section 5, the objective of the planner consists of two terms: 1) the discounted sum of the first $H$ rewards, and 2) the estimated long-term value of the penultimate step $s_H$ in the plan horizon, as determined by the agent’s $Q$ function.
The second part of the objective function is specifically included to ensure that the planner does not return myopic plans, and accounts for progress towards the long-horizon goal. Since the planner optimization objective includes this second term, even a small-horizon planner can output actions with proper awareness of long-horizon recovery events. The length of the horizon affects how close to the globally optimal solution the result is, with a tradeoff of computational cost.

To see this empirically, we reevaluated the double-gates+ environment (double integrator dynamics) with horizons of 1, 5, and 9. The graphs of attained reward and shield triggers from this experiment are attached as Figure 2 in the pdf. Comparing the H=1 and H=5 agents, the H=5 agent substantially outperforms the H=1 agent in both shield triggers and reward. Comparing H=5 and H=9, the H=9 agent reached high performance and low shield triggers faster than the H=5 agent. However, both the H=5 and H=9 agents converge to the same performance eventually.

**Experimental results**

We thank reviewer NRWz for suggesting the inclusion of PPO-Lagrangian as an additional baseline. We have evaluated it on all environments and appended the values to the tables of results. The new tables are attached to this response. Similar to CPO, PPO-lag was unable to make headway on the dynamic environments. We also correct bolding errors in the tables. In Table 1, DMPS, MPS, and REVEL are provably safe RL methods while CPO, PPO-lag, and TD3 are not. The table shows shield triggers for the former and safety violations for the latter.

**Concerns about assumption of perfect information and determinism**

The majority of prior work on PSRL assumes perfect information [1,2,3,4,5]. While traditional model-based RL methods can learn environment models even under partial observability/imperfect information, they cannot provably guarantee safety and are thus not suitable for tasks where safety violations are completely unacceptable.

Much prior work on provably safe locomotion also assumes determinism [3,4,5], with some such algorithms even having been deployed on real robots [3]. However, we agree with reviewers that extending our DMPS approach to stochastic environments is a promising direction for future work. In particular, since prior MPS work has been extended to work in stochastic settings [1,2], we believe our DMPS approach can be similarly extended to the stochastic setting as well.

[1] Anderson, G., Verma, A., Dillig, I., & Chaudhuri, S. (2020). Neurosymbolic Reinforcement Learning with Formally Verified Exploration. NeurIPS 2020

[2] Bastani, O., & Li, S. (2021). Safe Reinforcement Learning via Statistical Model Predictive Shielding. In D. A. Shell, M. Toussaint, & M. A. Hsieh (Eds.), RSS 2021.

[3] Vinod, A. P., Safaoui, S., Chakrabarty, A., Quirynen, R., Yoshikawa, N., & Cairano, S. D. (2022). Safe multi-agent motion planning via filtered reinforcement learning, ICRA 2022

[4] Zhang, W., & Bastani, O. (2019). MAMPS: Safe Multi-Agent Reinforcement Learning via Model Predictive Shielding. CoRR, abs/1910.12639.

[5] Zhu, H., Xiong, Z., Magill, S., & Jagannathan, S. (2019). An inductive synthesis framework for verifiable reinforcement learning, PLDI 2019

---

### Decision · Program_Chairs · 2024-09-25

**Decision:**

Accept (poster)

**Comment:**

The paper addresses provably safe RL problems where safety must be ensured even during training. In particular, it considers the optimal way to recover from potentially dangerous states with respect to the main objective.

The proposed approach is based on the Model Predictive Shielding, which follows a backup policy when the current policy proposes an unsafe action. The new method, called dynamic MPS (DMPS), integrates a local planner to select safe recovery actions that also aim to maximize the short and long-term rewards.

Although the reviewers consider that the method uses multiple known techniques, they also point out that the combination of such methods is novel. Moreover, they also praised the quality of the paper and the theoretical contribution related to the recovery regret. Furthermore, the rebuttal clarified some concerns regarding the computational overhead of the proposed method. Overall, the discussion raised some limitations of DMPS, which should be clarified in a new version of the paper.

I believe this paper makes a clear contribution to safe RL. Taking that into account and the unanimously positive recommendation of the reviewers, I recommend its acceptance.